# Active Representation Learning for General Task Space with Applications in Robotics

**Yifang Chen[1], Yingbing Huang[2], Simon S. Du[1*], Kevin Jamieson[1*], Guanya Shi[3*]**

[1] Paul G. Allen School of Computer Science & Engineering
University of Washington, Seattle,WA
`{yifangc, ssdu, jamieson, guanyas}@cs.washington.edu`

[2] University of Illinois Urbana-Champaign, Champaign, IL
`{yh21}@illinois.edu`

[3] Robotics Institute, Carnegie Mellon University, Pittsburgh, PA
`{{guanyas }@andrew.cmu.edu`
[*] Equal advising

## Abstract

Representation learning based on multi-task pretraining has become a powerful approach in many domains. In particular, task-aware representation learning aims to learn an optimal representation for a specific target task by sampling data from a set of source tasks, while task-agnostic representation learning seeks to learn a universal representation for a class of tasks. In this paper, we propose a general and versatile algorithmic and theoretic framework for *active representation learning*, where the learner optimally chooses which source tasks to sample from. This framework, along with a tractable meta algorithm, allows most arbitrary target and source task spaces (from discrete to continuous), covers both task-aware and task-agnostic settings, and is compatible with deep representation learning practices. We provide several instantiations under this framework, from bilinear and feature-based nonlinear to general nonlinear cases. In the bilinear case, by leveraging the non-uniform spectrum of the task representation and the calibrated source-target relevance, we prove that the sample complexity to achieve $\varepsilon$-excess risk on target scales with $(k^*)^2\|v^*\|_2^2\varepsilon^{-2}$ where $k^*$ is the effective dimension of the target and $\|v^*\|_2^2 \in (0, 1]$ represents the connection between source and target space. Compared to the passive one, this can save up to $\frac{1}{d_W}$ of sample complexity, where $d_W$ is the task space dimension. Finally, we demonstrate different instantiations of our meta algorithm in synthetic datasets and robotics problems, from pendulum simulations to real-world drone flight datasets. On average, our algorithms outperform baselines by $20\% - 70\%$. [1]

## 1 Introduction

Recently, few-shot machine learning has enjoyed significant attention and has become increasingly critical due to its ability to derive meaningful insights for target tasks that have minimal data, a scenario commonly encountered in real-world applications. This issue is especially prevalent in robotics where data collection and training data is prohibitive to collect or even non-reproducible (e.g., drone flying with complex aerodynamics [1] or legged robots on challenging terrains [2]). One

---

[1]Code in `https://github.com/cloudwaysX/ALMultiTask_Robotics`

37th Conference on Neural Information Processing Systems (NeurIPS 2023).

promising approach to leveraging the copious amount of data from a variety of other sources is multi-task learning, which is based on a key observation that different tasks may share a common low-dimensional representation. This process starts by pretraining a representation on source tasks and then fine-tuning the learned representation using a limited amount of target data ([3–7]).

In conventional supervised learning tasks, accessing a large amount of source data for multi-task representation learning may be easy, but processing and training on all that data can be costly. In real-world physical systems like robotics, this challenge is further amplified by two factors: (1) switching between different tasks or environments is often significantly more expensive (e.g., reset giant wind tunnels for drones [7]); (2) there are infinitely many environments to select from (i.e., environmental conditions are continuous physical parameters like wind speed). Therefore, it is crucial to minimize not only the number of samples, but the number of sampled source tasks, while still achieving the desired performance on the target task. Intuitively, not all source tasks are equally informative for learning a universally good representation or a target-specific representation. This is because source tasks can have a large degree of redundancy or be scarce in other parts of the task space. In line with this observation, Chen et al. [8] provided the first provable active representation learning method that improves training efficiency and reduces the cost of processing source data by prioritizing certain tasks during training with theoretical guarantees. On the other hand, many existing works [9–13] prove that it is statistically possible to learn a universally good representation by randomly sampling source tasks (i.e., the passive learning setting).

The previous theoretical work of [8] on active multi-task representation learning has three main limitations. First, it only focuses on a finite number of discrete tasks, treating each source independently, and therefore fails to leverage the connection between each task. This could be sub-optimal in many real-world systems like robotics for two reasons: (1) there are often infinitely many sources to sample from (e.g., wind speed for drones); (2) task spaces are often highly correlated (e.g., perturbing the wind speed will not drastically change the aerodynamics). In our paper, by considering a more general setting where tasks are parameterized in a vector space $\mathcal{W}$, we can more effectively leverage similarities between tasks compared to treating them as simply discrete and different. Secondly, the previous work only considers a single target, while we propose an algorithm that works for an arbitrary target space and distribution. This is particularly useful when the testing scenario is time-variant. Thirdly, we also consider the task-agnostic setting by selecting $\mathcal{O}(k)$ representative tasks among the $d_W$ high dimension task space, where $k \ll d_W$ is the dimension of the shared representation. Although this result does not improve the total source sample complexity compared to the passive learning result in the bilinear setting [12], it reduces the number of tasks used in the training and therefore implicitly facilitates the training process.

In addition to those theoretical contributions, we extend our proposed algorithmic framework beyond a pure bilinear representation function, including the *known* nonlinear feature operator with unknown linear representation (e.g., random features with unknown coefficients), and the totally *unknown nonlinear representation* (e.g., deep neural network representation). While some prior works have considered nonlinear representations [9, 10, 14, 13] in passive learning, the studies in active learning are relatively limited [8]. All of these works only consider non-linearity regarding the input, rather than the task parameter. In this paper, we model task-parameter-wise non-linearity and show its effectiveness in experiments. Note that it particularly matters for task selections because the mapping from the representation space to task parameters to is no longer linear.

See more related works and how our problem scope is different from theirs in Appendix A.

## 1.1 Summery of contributions

- We propose the first generic active representation learning framework that admits any arbitrary source and target task space. This result greatly generalizes previous works where tasks lie in the discrete space and only a single target is allowed. To show its flexibility, we also provide discussions on how our framework can accommodate various supervised training oracles and optimal design oracles. (Section 3)

- We provide theoretical guarantees under a benign setting, where inputs are i.i.d. and a unit ball is contained in the overall task space, as a compliment to the previous work where tasks lie on the vertices of the whole space. In the target-aware setting, to identify an $\varepsilon$-good model our method requires a sample complexity of $\widetilde{\mathcal{O}}(k d_X (k^*)^2 \|v^*\|_2^2 \min\{k^*, \kappa^2\} \varepsilon^{-2})$ where $k^*$ is the effective dimension of the target, $\kappa$ is the conditional number of representation matrix, and

$\|v^*\|_2^2 \in (0, 1]$ represents the connection between source and target space that will be specified in the main paper. Compared to passive learning, our result saves up to a factor of $\frac{k^2}{d_W}$ in the sample complexity when targets are uniformly spread over the $k$-dim space and up to $\frac{1}{d_W}$ when targets are highly concentrated. Our results further indicate the necessity of considering the continuous space by showing that directly applying the previous algorithm onto some discretized sources in the continuous space (e.g., orthonormal basis) can lead to worse result. Finally, ignoring the tasks used in the warm-up phases, in which only a few samples are required, both the target-aware and the target-agnostic cases can save up to $\widetilde{\mathcal{O}}(k^* + k)$ number of tasks compared to the passive one which usually requires $d_W$ number of tasks. (Section 4)

- We provide comprehensive experimental results under different instantiations beyond the benign theoretical setting, studying synthetic and real-world scenarios: 1) For the synthetic data setting in a continuous space, we provide results for pure linear, known nonlinear feature operator $\psi_X$ and unknown nonlinear representation $\phi_X$. Our target-aware active learning (AL) approach shows up to a significant budget saving (up to $68\%$) compared to the passive approach and the target-agnostic AL approach also shows an advantage in the first two cases. 2) In a pendulum simulation with continuous task space, we provide the results for known nonlinear feature operator $\psi_X$ and $\psi_W$ and show that our target-aware AL approach has up to $20\%$ loss reduction compared to the passive one, which also translates to better nonlinear control performance. 3) Finally, in the real-world drone dataset with a discrete task space, we provide results for unknown linear and nonlinear representation $\phi_X$ and show that our target-aware AL approach converges much faster than the passive one. (Section 5)

## 2 Preliminary

**Multi-task (or multi-environments).** Each task or environment is parameterized by a known vector $w \in \mathbb{R}^{d_W}$. We denote the source and target task parameter space as $\mathcal{W}_{\text{source}} \subset \mathbb{R}^{d_W}, \mathcal{W}_{\text{target}} \subset \mathbb{R}^{d_W}$. These spaces need not be the same (e.g., they could be different sub-spaces). In the discrete case, we set $w$ as a one-hot encoded vector and therefore we have in total $d_W$ number of candidate tasks while in the continuous space, there exist infinitely many tasks. For convenience, we also use $w$ as the subscript to index certain tasks. In addition, we use $\nu_{\text{source}} \in \Delta(\mathcal{W}_{\text{source}}), \nu_{\text{target}} \in \Delta(\mathcal{W}_{\text{target}})$ to denote the task distribution for the sources and targets.

**Data generation.** Let $\mathcal{X} \in \mathbb{R}^{d_X}$ be the input space. We first assume there exists some *known* feature/augmentation operator $\psi_X : \mathcal{X} \to \mathbb{R}^{d_{\psi_X} \geq d_W}, \psi_W : \mathcal{W} \to \mathbb{R}^{d_{\psi_W} \geq d_W}$, that can be some nonlinear operator that lifts $w, x$ to some higher dimensional space (e.g., random Fourier features [15]). Notice that the existence of non-identical $\psi$ indicates the features are not pairwise independent and the design space of $\mathcal{W}_{\text{source}}$ is not benign (e.g., non-convex), which adds extra difficulty to this problem.

Then we assume there exists some *unknown* underlying representation function $\phi_X : \psi(\mathcal{X}) \to \mathcal{R}$ which maps the augmented input space $\psi(\mathcal{X})$ to a shared representation space $\mathcal{R} \in \mathbb{R}^k$ where $k \ll d_{\psi_X}, k \leq d_{\psi_W}$, and its task counterparts $\phi_W : \psi(\mathcal{W}) \to \mathcal{R}$ which maps parameterized task space to the feature space. Here the representation functions are restricted to be in some function classes $\Phi$, e.g., linear functions, deep neural networks, etc.

In this paper, we further assume that $\phi_W$ is a linear function $B_W \in \mathbb{R}^{k \times d_{\psi_W}}$. To be more specific, for any fixed task $w$, we assume each sample $(x, y) \sim \nu_w$ satisfies

$$y = \phi_X(\psi_X(x))^\top B_W \psi_W(w) + \xi, \quad \xi \sim \mathcal{N}(0, \sigma^2) \tag{1}$$

For convenience, we denote $Z_w$ as the collection of $n_w$ sampled data $(x_w^1, y_w^1), ..., (x_w^{n_w}, y_w^{n_w}) \sim \mu_w$. We note that when $\psi_X, \psi_W$ is identity and $\phi_X$ is linear, this is reduced to standard linear setting in many previous papers [9, 11, 12, 8].

**The task diversity assumption.** There exists some distribution $p \in \Delta(\mathcal{W}_{\text{source}})$ that $\mathbb{E}_{w \sim p} \lambda_{\min}(B_W \psi_W(w) \psi_W(w)^\top B_W^\top) > 0$, which suggests the source tasks are diverse enough to learn the representation.

**Data collection protocol.** We assume there exists some i.i.d. data sampling oracle given the environment and the budget. To learn a proper representation, we are allowed access to an *unlimited* $n_{\text{source}}$ number of data from source tasks during the learning process by using such an oracle. Then at

the end of the algorithm, we are given a few-shot of *mix* target data $Z_{\text{target}} = \{Z_w\}_{w \sim \nu_{\text{target}}}$ which is used for fine-tuning based on learned representation $\hat{\phi}_X$. Denote $n_{\text{target}}$ as the number of data points in $Z_{\text{target}}$.

**Data collection protocol for target-aware setting.** When the target task is not a singleton, we additionally assume a few-shot of *known environment* target data $\dot{Z}_{\text{target}} := \{Z_w, w\}_{w \in \dot{W}_{\text{target}}}$, where $|\dot{W}_{\text{target}}| = \dim(\mathcal{W}_{\text{target}})$ and $\dot{W}_{\text{target}} = \{\arg\max_{W \in \mathcal{W}_{\text{target}}} \lambda_{\min}(WW^\top)\}$. Again denote $\dot{n}_{\text{target}}$ as the number of data points in $\dot{Z}_{\text{target}}$, we have $\dot{n}_{\text{target}} \approx n_{\text{target}}^{2/3} \ll n_{\text{source}}$.

*Remark* 2.1. Here $|\dot{W}_{\text{target}}|$ represents vectors that can cover every directions of $\mathcal{W}_{\text{target}}$ space. This extra $\dot{Z}_{\text{target}}$ requirement comes from the non-linearity of $l_2$ loss and the need to learn the relationship between sources and targets. We want to emphasize that such an assumption implicitly exists in previous active representation learning [8] since $\dot{Z}_{\text{target}} = Z_{\text{target}}$ in their single target setting. Nevertheless, in a passive learning setting, only mixed $\dot{Z}_{\text{target}}$ is required since no source selection process involves. Whether such a requirement is necessary for target-aware active learning remains an open problem.

**Other notations.** Let $e_i$ to be one-hot vector with 1 at $i$-th coordinates and let $\epsilon_i = 2^{-i}$.

## 2.1 Goals

**Expected excess risk.** For any target task space $\mathcal{W}_{\text{target}}$ and its distribution $\nu_{\text{target}}$ over the space, as well as a few-shot examples as stated in section 2, our goal is to minimize the expected excess risk with our estimated $\hat{\phi}_X$

$$\text{ER}(\hat{\phi}_X, \nu_{\text{target}}) = \mathbb{E}_{w_0 \sim \nu_{\text{target}}} \mathbb{E}_{(x,y) \sim \nu_{w_0}} \|\hat{\phi}_X(\psi_X(x))^\top \hat{w}_{\text{avg}} - y\|_2$$

where $\hat{w}_{\text{avg}} = \arg\min_w \sum_{(x,y) \in Z_{\text{target}}} \|\hat{\phi}_X(\psi_X(x))w - y\|_2$, which average model estimation that captures the data behavior under the expected target distribution. Note that the $\mathcal{W}_{\text{target}}, \nu_{\text{target}}$ are given in advance in the target-aware setting.

**The number of tasks.** Another side goal is to save the number of long-term tasks we are going to sample during the learning process. Since a uniform exploration over $d_W^{\text{source}}$-dimension is unavoidable during the warm-up stage, we define long-term task number as

$$\left| \left\{ w \in \mathcal{W}_{\text{source}} \mid n_w \geq \tilde{\Omega}(\varepsilon^{-\alpha}) \right\} \right|$$

where $\alpha$ is some arbitrary exponent and $\varepsilon$ is the target accuracy and $n_w$ is number of samples sampled from task $w$ as defined above.

## 3 A general framework

Our algorithm 1 iteratively estimates the shared representation $\hat{\phi}_X, \hat{B}_W$ and the next target relevant source tasks which the learner should sample from by solving several optimal design oracles

$$g(f, A) = \min_{q \in \Delta(\mathcal{W}_{source})} \lambda_{\max} \left( (\int q(w) f(w) f(w)^\top)^{-1} A \right) \tag{2}$$

This exploration and exploitation (target-aware exploration here) trade-off is inspired by the classical $\epsilon$-greedy strategy, but the key difficulty in our work is to combine that with multi-task representation learning and different optimal design problems. The algorithm can be generally divided into three parts, and some parts can be skipped depending on the structure and the goal of the problem.

- **Coarse exploration:** The learner uniformly explores all the directions of the $\mathcal{W}_{\text{source}}$ (denoted by distribution $q_0$) in order to find an initial $k$-dimension subspace $V$ that well spans over the representation space (i.e., $\frac{1}{c} B_W B_W^\top \leq B_W VV^\top B_W^\top \leq c B_W B_W^\top$ for some arbitrary constant $c \leq \frac{d_{\psi_W}}{k}$). To give an intuitive example, suppose $B_W \in \mathbb{R}^{2 \times d_{\psi_W}^{\text{source}}+1}$ has the first half column equals $e_1$ and the second hard equals $e_2$. Then instead of uniformly choosing $\{e_i\}_{i \in [d_{\psi_W}^{\text{source}}]}$ task, we only need explore over two tasks $V[1] = \sqrt{\frac{2}{d_{\psi_W}^{\text{source}}}}[1, 1, \ldots, 0, 0, \ldots], V[2] = \sqrt{\frac{2}{d_W^{\text{source}}}}[0, 0, \ldots, 1, 1, \ldots]$.

---

**Algorithm 1** Active multi-task representation learning (general templates)

---

1: **Inputs:** Candidate source set $\mathcal{W}_{\text{source}}$. Classes of candidate representation function $\Phi_X, \Phi_W$ and the known feature operator $\psi_X, \psi_W$.

2: **[Target-aware only] Inputs:** Target set $\mathcal{W}_{\text{target}}$ and distribution $\nu_{\text{target}}$. Few-shot sample $\dot{Z}_{\text{target}}$ as defined in the preliminary.

3: Stage 1: Coarse exploration. (Warm-up stage)

4: Set initial sampling distribution $q_0 = g(\psi_W, I_{d_{\psi_W}})$ where $g$ is defined in Eqn. 2

5: Set $n_0 \approx \text{poly}(d_{\psi_X}, k) + \text{poly}(d_{\psi_W}, k)$. Collect $n_0 q_0(w)$ data for each task denoted as $\{Z_w\}_{w|q_0(w) \neq 0}$ and update $\hat{\phi}_X \leftarrow \mathcal{O}^X_{\text{offline } 0}(\{Z_w\}_{w|q_0(w) \neq 0}, \psi_X)$ and $\hat{B}_W \leftarrow \mathcal{O}^W_{\text{offline}}(\{Z_w\}_{w|q_0(w) \neq 0}, \hat{\phi}_X)$

6: **for** $j = 1, 2, 3, \ldots$ **do**

7:     Stage 2: Fine target-agnostic exploration (Directly choose $q_1^j = q_0$ when $k = \Theta(d_W)$)

8:     Compute the exploration sampling distribution $q_1^j = g(\hat{B}_W \circ \psi_W, I_k)$

9:     $n_1^j \approx \text{poly}(d_{\psi_X}, k)\epsilon_j^{-\frac{4}{3}}$. Collect $n_1^j q_1^j(w)$ data for each task denoted as $\{Z_w\}_{w|q(w) \neq 0}$ and update $\hat{\phi}_X \leftarrow \mathcal{O}^X_{\text{offline } 1}(\{Z_w\}_{w|q_1^j(w) \neq 0}, \psi_X)$ and $\hat{B}_W \leftarrow \mathcal{O}^W_{\text{offline}}(\{Z_w\}_{w|q_1^j(w) \neq 0}, \dot{Z}_{\text{target}}, \hat{\phi}_X)$

10:     **[Target-aware only]** Stage 3: Fine target-aware exploration

11:     Compute the exploitation sampling distribution $q_2^j = g(\hat{B}_W \circ \psi_W, \Sigma_{\text{regu}})$ where $\Sigma_{\text{regu}}$ is the regularized version of $\hat{B}_W(\mathbb{E}_{w_0 \sim \nu_0} w_0 w_0^\top)\hat{B}_W^\top$ after clipping out insignificant eigenvalues.

12:     Set $n_2^j \approx \text{poly}(d_{\psi_X}, k)\epsilon_j^{-2}$. Collect $n_2^j q_2^j(w)$ data for each task denoted as $\{Z_w\}_{w|q_2^j(w) \neq 0}$ and update $\hat{\phi}_X \leftarrow \mathcal{O}^X_{\text{offline } 3}(\{Z_w\}_{w|q_1^j(w) \neq 0 \text{ and } q_2^j(w) \neq 0}, \psi_X)$.

13: **end for**

14: **Return** $\hat{\phi}_X$

---

We want to highlight that the sample complexity of this warm-up stage only scales with $d_{\psi_X}, k$ and the spectrum-related parameters of $B_W$ (i.e., $\kappa(B_W), \sigma_{\min}(B_X)$), not the desired accuracy $\varepsilon$.

- **Fine target-agnostic exploration:** The learner iteratively updates the estimation of $V$ and uniformly explore for $\widetilde{\mathcal{O}}(\epsilon_j^{-\frac{4}{3}})$ times on this $k$, instead of $d_{\psi_W}$ subspace, denoted by distribution $q_1$. (Note this $\epsilon_j^{-\frac{4}{3}}$ comes from the exploration part in $\epsilon$-greedy, which is $(n_2^j)^{\frac{2}{3}}$) Such reduction not only saves the cost of maintaining a large amount of physical environment in real-world experiments but also simplifies the non-convex multi-task optimization problem. Of course, when $k = \Theta(d_{\psi_W})$, we can always uniformly explore the whole ($d_{\psi_W}$ space as denoted in the algorithm. Note that theoretically, $q_1$ only needs to be computed once as shown in 4. In practice, to further improve the accuracy while saving the task number, the $q_1$ can be updated only when a significant change from the previous one happens, which is adopted in our experiments as shown in appendix E.1.

- **Fine target-aware exploration.** In the task-awareness setting, the learner estimates the most-target-related sources parameterized by $\{w\}$ based on the current representation estimation and allocates more budget on those, denoted by distribution $q_2$. By definition, $q_2$ should be more sparse than $q_1$ and thus allowing the final sample complexity only scales with $k^*$, which measures the effective dimension in the source space that is target-relevant.

**Computational oracle for optimal design problem.** Depending on the geometry of $\{\psi_W(w)\}_{w \in \mathcal{W}_{\text{source}}}$, the learner should choose proper offline optimal design algorithms to solve $g(f, A)$. Here we propose several common choices. 1). When $\mathcal{W}_{\text{source}}$ contains a ball, we can approximate the solution via an eigendecomposition-based closed-form solution with an efficient projection as detailed in Section 4. 2) When $\mathcal{W}_{\text{source}}$ is some other convex geometry, we can approximate the result via the Frank-Wolfe type algorithms [16], which avoids explicitly looping over the infinite task space. 3) For other even harder geometry, we can use discretization or adaptive sampling-based approximation [17]. In our experiments, we adopt the latter one and found out that its running time cost is almost neglectable in our pendulum simulator experiment in Section 5, where the $\psi_W$ is a polynomial augmentation.

**Offline optimization oracle $\mathcal{O}^X_{\text{offline}}$.** Although we are in the continuous setting, the sampling distribution $q_0, q_1, q_2$ is sparse. Therefore, our algorithm allows any proper passive multi-task

learning algorithm, either theoretical or heuristic one, to plugin the $\mathcal{O}_{\text{offline}}^X$. Some common choices include gradient-based joint training approaches[18–21], the general non-convex ERM [9] and other more carefully designed algorithms [12, 22]. We implement the first one in our experiments (Section 5) to tackle the nonlinear $\psi_X, \phi_X$ and give more detailed descriptions of the latter two in Section 4 and Appendix B.1 to tackle the bilinear model.

# 4 A theoretical analysis under the benign $\mathcal{W}_{\text{source}}$ setting

## 4.1 Assumptions

**Assumption 4.1** (Geometry of the task space). *We assume the source task space $\mathcal{W}_{source}$ is a unit ball $\mathbb{B}^{d_W^{source}}(1)$ that span over the first $d_W^{source} \geq \frac{1}{2}d_W$ without loss of generality, while the target task space $\mathcal{W}_{target} \subset \mathbb{R}^{d_W}$ can be any arbitrary $\mathbb{B}^{d_W^{target}}(1)$.*

Under this assumption, we let $B_W^{\text{source}}$ denote the first $d_W^{\text{source}}$ columns of $B_W$, which stands for the source-related part of $B_W$. And $B_W^{\text{target}}$

Then we assume the bilinear model where $\phi_X = B_X \in \mathbb{B}^{d_X \times k}$ and $\psi_X, \psi_W = I$. Therefore, $d_{\psi_X} = d_X, d_{\psi_W} = d_W$. Moreover the model satisfies the following assumptions

**Assumption 4.2** (Benign $B_X, B_W$). *$B_X$ is an orthonormal matrix. Each column of $B_W$ has magnitude $\Theta(1)$ and $\sigma_{\min}(B_W^{source}) > 1$. Suppose we know $\bar{\kappa} \geq \kappa(B_W^{source}), \sigma_{\max}(B_W^{target})$ and $\underline{\sigma} \leq \sigma_{min}(B_W^{source}), \sigma_{min}(B_W^{target})$. Trivially, $\bar{\kappa} = \sqrt{d_W}, \underline{\sigma} = 1$.*

Finally, the following assumption is required since we are using a training algorithm in [12] and might be able to relax to sub-gaussian by using other suboptimal oracles.

**Assumption 4.3** (Isotropic Gaussian Input). *For each task $w$, its input $i$ satisfies $x_{i,w} \sim \mathcal{N}(0, I_d)$.*

## 4.2 Algorithm

Here we provide the target-aware theory and postpone the target-agnostic in the Appendix. C since its analysis is covered by the target-aware setting.

This target-aware algorithm 2 follows the 3-stage which corresponds to sampling distribution $q_0, q_1, q_2$ with explicit solutions. Notice that calculating $q_1$ once is enough for theoretical guarantees.

We use existing passive multi-task training algorithms as oracles for $\mathcal{O}_{\text{offline }1}^X, \mathcal{O}_{\text{offline }2}^X$ and use the simple ERM methods for $\mathcal{O}_{\text{offline}}^W$ based on the learned $\hat{B}$. For the coarse exploration and fine target-agnostic exploration stage, the main purpose is to have a universal good estimation in all directions of $B_X$. ( i.e., upper bound the $\sin(\hat{B}_X, B_X)$) Therefore we choose the alternating minimization (MLLAM) proposed in [12]. On the other hand, for the fine target-aware exploration, we mainly care about final transfer learning performance on learned representation. Therefore, we use a non-convex ERM from [9]. We defer the details and its theoretical guarantees for $\mathcal{O}_{\text{offline}}$ into Appendix B.1.

Note the major disadvantage from [9] comes from its sample complexity scaling with a number of training source tasks, which will not be a problem here since in $\mathcal{O}_{\text{offline, 3}}^X$ since only $k + k^* \ll d_W$ number of tasks are used. The major benefit of using non-convex ERM comes from its generality that it works even for the non-linear setting and is not tied with a specific algorithm. That is to say, as long as there exists other theoretical or heuristic oracles $\mathcal{O}_{\text{offline, 1}}^X, \mathcal{O}_{\text{offline, 2}}^X$ giving a similar guarantee, stage 3 always works.

## 4.3 Results

**Theorem 4.1** (Informal). *By running Algo. 2, in order to let $ER(\hat{\phi}_X, \nu_{target}) \leq \varepsilon^2$ with probability $1 - \delta$, the number of source samples $n_{source}$ is at most*

$$\widetilde{\mathcal{O}}\left((kd_X + \log(1/\delta))(k^*)^2 \min\{k^*, \kappa^2(B_W)\} \max_i \|W_i^*\|_2^2 \varepsilon^{-2} + \textit{low-order}\right)$$

*Here $k^* = rank(\mathbb{E}_{w_0 \sim \nu_{target}} B_W w_0 w_0^\top B_W^\top)$ represents the effective dimension of target and*

$$W_i^* = \arg\min_{w \in \mathcal{W}_{source}} \|w\|_2 \quad s.t \quad B_W^{source} w = u_i \sqrt{\lambda_i} \text{ where } U, \Lambda \leftarrow \textit{Eig}(\mathbb{E}_{w_0 \sim \nu_{target}} B_W w_0 w_0^\top B_W^\top).$$

---

**Algorithm 2** Target-aware algorithm for benign source space

---

1: **Inputs:** Target probability $\delta$, $\bar{\kappa}$, $\underline{\sigma}$. Some constant $\beta_1, \beta_2, \beta_3$. Others same as Algo. 1.
2: Set $q_0$ as $q_0(e_t) = \frac{1}{d_W}, \forall t \in d_W$, and $q_0(w) = 0$ otherwise
3: Set $n_0 = \beta_1 \bar{\kappa}^2 \left( k^3 d_X \bar{\kappa}^2 + d_W^{\frac{3}{2}} \underline{\sigma}^{-2} \sqrt{k + \log(1/\delta)} \right)$. Collect $n_0 q_0(w)$ data for each task denoted as $\{Z_w\}_{w|q(w)\neq 0}$
4: Update $\hat{B}_X \leftarrow \mathcal{O}_{\text{offline 1}}^X(\{Z_w\}_{w|q_0(w)\neq 0})$ and $\hat{B}_W^{\text{source}} \leftarrow \mathcal{O}_{\text{offline}}^W(\{Z_w\}_{w|q_0(w)\neq 0}, \hat{B}_X)$
5: Compute $q_1$ as $q_1(v_i) = \frac{1}{k}, \forall i \in k$, and $q_0(w) = 0$ otherwise. Here $v_i$ is the $i$-th vector of $V$, where $U, D, V \leftarrow \text{SVD}(\hat{B}_W^{\text{source}})$
6: **for** $j = 1, 2, 3, \dots$ **do**
7:     Set $n_1^j = \beta_2 \epsilon_j^{-\frac{4}{3}} k^{\frac{5}{3}} d_W^{\frac{2}{3}} d_X^{\frac{1}{3}} \left( k^{\frac{2}{3}} d_W^{\frac{1}{3}} \underline{\sigma}^{-\frac{4}{3}} + \bar{\kappa}^2 \underline{\sigma}^{-\frac{2}{3}} \right)$. Collect $n_1^j q_1(w)$ data for each task denoted as $\{Z_w\}_{w|q_1(w)\neq 0}$.
8:     Update $\hat{B}_X \leftarrow \mathcal{O}_{\text{offline 2}}^X(\{Z_w\}_{w|q_1(w)\neq 0})$, $\hat{B}_W^{\text{source}} \leftarrow \mathcal{O}_{\text{offline}}^W(\{Z_w\}_{w|q_1(w)\neq 0}, \hat{B}_X)$
        and $\hat{B}_W^{\text{target}} \leftarrow \mathcal{O}_{\text{offline}}^W(\dot{Z}_{\text{target}}, \hat{B}_X)$
9:     Find a set of target-aware tasks parameterized by $\tilde{W}_j$ with each column $i$ as

$$\tilde{W}_j(i) = \text{Proj}_{\mathcal{W}_{\text{source}}} w_i' = \frac{w_i'}{\|w_i'\|_2}$$

$$\text{where } w_i' = \arg\min_w \|w\|_2 \quad \text{s.t.} \quad \hat{B}_{W,j}^{\text{source}} w = u_i \sqrt{\lambda_i} \quad \forall \Lambda_i \geq 8(kd_W)^{\frac{3}{2}} \sqrt{\frac{d_X}{n_1}}$$

$$\text{where } U, \Lambda \leftarrow \text{Eig} \left( \mathbb{E}_{w_0 \sim \nu_{\text{target}}} \left[ \hat{B}_{W,j}^{\text{target}} w_0 (\hat{B}_{W,j}^{\text{target}} w_0)^\top \right] \right)$$

10:     Compute $q_2^j$ as $q_2^j(w) = \frac{1}{\# \text{col}(\tilde{W}_j)}, \forall w \in \text{col}(\tilde{W}_j)$ and $q_2^j(w) = 0$ otherwise
11:     Assign $n_2^j$ total sampling budget as $\# \text{col}(\tilde{W}_j) \beta_3 \max_i \|W_j'(i)\|_2^2 \epsilon_j^{-2}$
12:     Collect $n_2^j(w) = n_2^j q_2^j(w)$ data for each task denoted as $\{Z_w\}_{w|q_2(w)\neq 0}$.
13:     Update the model, note that both data collected from stage 2 and stage 3 are used.

$$\tilde{B}_X \leftarrow \mathcal{O}_{\text{offline 3}}^X(\{Z_w\}_{w|q_1(w)\neq 0 \text{ and } q_2(w)\neq 0})$$

14: **end for**
15: **Return** $\tilde{B}_X$

---

*As long as the number of target samples satisfies*

$$n_{\text{target}} \geq \tilde{\Omega}((k + \log(1/\delta))\varepsilon^{-2}), \quad \dot{n}_{\text{target}} \gtrsim \tilde{\Omega} \left( \varepsilon^{-\frac{4}{3}} (k^*)^{\frac{2}{3}} \sqrt{k} \left( d_W^{\frac{1}{2}} \underline{\sigma}^{-\frac{4}{3}} + k^{-\frac{2}{3}} d_W^{\frac{1}{6}} \bar{\kappa}^2 \underline{\sigma}^{-\frac{1}{3}} \right) \right)$$

**Comparison with passive learning.** By choosing $\{e_i\}_{i \in [d_W^{\text{source}}]}$ as a fixed source set, we reduce the problem to a discrete setting and compare it with the passive learning. In [9], the authors get $N_{\text{total}}$ as most $\frac{kd_X d_W \|\mathbb{E}_{w_0 \sim \nu_{\text{target}}} B_W w_0 w_0^\top B_W^\top\|}{\sigma_{\min}^2(B_W^{\text{source}})} \varepsilon^{-2}$. We first consider the cases in their paper that the target task is uniformly spread $\|\mathbb{E}_{w_0 \sim \nu_{\text{target}}} B_W w_0 w_0^\top B_W^\top\| = \frac{1}{k}$.

- When the task representation is well-conditioned $\sigma_{\min}^2(B_W^{\text{source}}) = \frac{d_W}{k}$. We have a passive one as $\tilde{\mathcal{O}}(kd_X \varepsilon^{-2})$ while the active one $\tilde{\mathcal{O}}(kd_X \frac{k^2}{d_W} \varepsilon^{-2})$ (See Lemma B.8 for details), which suggests as long as $d_W \gg k^2$, our active learning algorithm gain advantage even in a relatively uniform spread data and representation conditions.

- Otherwise, we consider the extreme case that $\sigma_{\min}^2(B_W^{\text{source}}) = 1$. We have passive one $\tilde{\mathcal{O}}(d_X d_W \varepsilon^{-2})$ while the active one $\tilde{\mathcal{O}}(k^3 d_X \varepsilon^{-2})$. Notice here we require $d_W \gg k^3$.

Both of them indicate the necessity of considering the continuous case with large $d_W$ even if everything is uniformly spread. On the other hand, whether we can achieve the same result as the passive one when $d_W \leq k^3$ remains to be explored in the future.

We then consider the single target $w_0$ case.

- With well-conditioned $B_W$, the passive one now has sample complexity $\mathcal{O}(k^2 d_X \varepsilon^{-2})$ while the active gives a strictly improvement $\mathcal{O}(\frac{k^2 d_X}{d_W} \varepsilon^{-2})$.

- With ill-conditioned $B_W$ where $\sigma_{\min}(B_W) = 1$ and $\max_i \|W_i^*\| = 1$, that is, only a particular direction in source space contributes to the target. The Passive one now has sample complexity $\mathcal{O}(k d_X d_W \varepsilon^{-2})$ while our active one only has $k d_X \varepsilon^{-2}$, which demonstrates the benefits of our algorithm in unevenly distributed source space.

**Comparison with previous active learning.** By using the same discrete reduction and set single target $w_0$, we compare our result with the current state-of-art active representation algorithm in [23]. They achieves $\widetilde{\mathcal{O}}(k d_X \|\nu\|_1^2 \varepsilon^{-2})$, where $\nu = \arg\min_\nu \|\nu\|_1$ s.t $B_W \nu = B_W w_0$. On the other hand, our active one gives $\widetilde{\mathcal{O}}(k d_X \|w^*\|_2^2 \varepsilon^{-2})$, where $w^* = \arg\min_\nu \|\nu\|_2$ s.t $B_W \nu = B_W w_0$, which is strictly better than the discrete one. This again indicates the separation between continuous and discrete cases where in fixed discrete sets, the $L_1$ norm regularization is strictly better than $L_2$.

Furthermore, when a fixed discrete set is given, which is exactly the setting in [23]. Their algorithm can be seen as a computationally efficient reduction under ours.(Appendix B.5.)

**Save task number.** When ignoring the short-term initial warm-up stage, we only require maintaining $\widetilde{\mathcal{O}}(k + \log(N_{\text{total}} k^*))$ number of source tasks, where the first term comes from $q_1$ in the target-agnostic stage and the second term comes from $q_2$ in the target-aware stage.

## 5 Experiment

In this section, we provide experimental results under different instantiations of the Algorithm 1, and all of them show the effectiveness of our strategy both in target-aware and target-agnostic settings.

### 5.1 Settings

**Datasets and problem definition.** Our results cover the different combinations of $\psi_X, \phi_X, \psi_W$ as shown in Table 1. Here we provide a brief introduction for the three datasets and postpone the details into Appendix E. [2]

|  | identity $\psi_W$ | nonlinear $\psi_W$ |
|---|---|---|
| identity $\psi_X$ and linear $\phi_X$ | synthetic, drone | NA |
| nonlinear $\psi_X$ and linear $\phi_X$ | synthetic | pendulum simulator |
| identity $\psi_X$ and nonlinear $\phi_X$ | synthetic, drone | NA |

Table 1: Summary of different instantiations

- **Synthetic data.** We generate data that strictly adhere to our data-generating assumptions and use the same architecture for learning and predicting. When $\phi_X$ is nonlinear, we use a neural network $\phi_X$ to generate data and use a slightly larger neural net for learning. The goal for synthetic data is to better illustrate our algorithm as well as serve as the first step to extend our algorithm on various existing datasets.

- **Pendulum simulator.** To demonstrate our algorithm in the continuous space. we adopt the multi-environment pendulum model in [24] and the goal is to learn a $w$-dependent residual dynamics model $f(x, w) \in \mathbb{R}$ where $x$ is the pendulum state and $w \in \mathbb{R}^5$ including external wind, gravity and damping coefficients. $f(x, w)$ is highly nonlinear with respective to $x$ and $w$. Therefore we use known non-linear feature operators $\psi_X, \psi_W$. In other words, this setting can be regarded as a misspecified linear model. It is also worth noting that due to the non-invertibility of $\psi_W$, the explicit selection of a source via a closed form is challenging. Instead, we resort to an adaptive sampling-based method discussed in Section 3. Specifically, we uniformly sample $w$ from the source space, select the best $w'$, and then uniformly sample around this $w'$ at a finer grain. Our findings indicate that about 5 iterations are sufficient to approximate the most relevant source.

- **Real-world drone flight dataset [7].** The Neural-Fly dataset [7] includes real flight trajectories using two different drones in various wind conditions. The objective is to learn the residual

---

[2]Github Link: https://github.com/cloudwaysX/ALMultiTask_Robotics

aerodynamics model $f(x, w) \in \mathbb{R}^3$ where $x \in \mathbb{R}^{11}$ is the drone state (including velocity, attitude, and motor speed) and $w$ is the environment condition (including drone types and wind conditions). We collect 6 different $w$ and treat each dimension of $f(x, w)$ as a separate task. Therefore $w$ is reformulated as a one-hot encoded vector in $\mathbb{R}^{18}$.

For each dataset/problem, we can choose different targets. For simplicity, in the following subsection, we present results for one target task for each problem with 10 random seeds regarding random data generation and training, and put more results in Appendix E. In all the experiments, we use a gradient-descent joint training oracle, which is a standard approach in representation learning.

## 5.2 Results

Those results encapsulate the effectiveness of active learning in terms of budget utilization and test loss reduction. In the drone dataset, we further demonstrate its ability in identifying relevant source tasks (see Figure 2). We note that in two robotics problems (pendulum simulation and real-world drone dataset), the active learning objective is to learn *a better dynamics model*. However, in the pendulum simulation, we deploy a model-based nonlinear controller which translates better dynamics modeling to enhanced control performance (see Figure 1 and Appendix E.2).

|  | Target-aware AL | Target-agnostic AL |
|---|---|---|
| identity $\psi_X$ and linear $\phi_X$ | 38.7% | 51.6% |
| nonlinear $\psi_X$ and linear $\phi_X$ | 38.7% | 45.2% |
| identity $\psi_X$ and non-linear $\phi_X$ | 32.0% | 68.0% |

Table 2: Results on synthetic data. Using the test loss of the final output model from passive learning as a baseline, we show the ratio between the budget required by target-aware/target-agnostic active learning to achieve a similar loss and the budget required by passive learning.

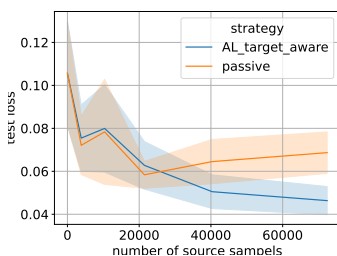 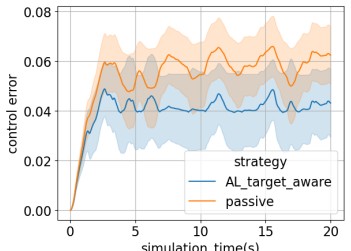

Figure 1: Results on pendulum simulator for a specific target. **Left:** The test loss of the estimated model $\hat{f}$. The passive strategy suffers from negative transfer while the active strategy steadily decreases. **Right:** The control error using final output $\hat{f}$. Here we use a model-based nonlinear policy $\pi(x, \hat{f})$. The model learned from active strategy leads to better control performance.

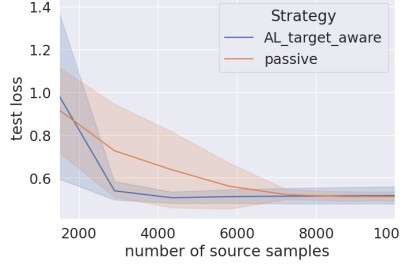 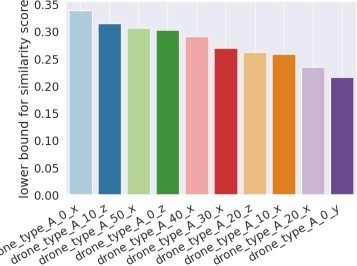

Figure 2: Results on the real drone dataset [7] with target `drone_type_A_30_z`. Source data includes two drone types A and B, six wind speeds from 0 to 50, and three directions x-y-z. We present results for linear $\phi_X$ here and postpone the non-linear $\phi_X$ case in Appendix E.3. **Left:** The test loss of the estimated bilinear model $\hat{f}$. The passive strategy converges slower than the active strategy. **Right:** Top 10 the most similar source tasks. Given the target environment, the algorithm successfully finds the other `drone_type_A` environments as relevant sources. See more explanations in Appendix E.3.

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
