# Contents

# A Related works

Here we give a brief summary of other representation learning or multi-task papers that are related but different in some aspects

**Multi-task with negative correlation**    Some multi-task works [25–27, 22] assume different tasks don't share the same representation, so learning on one task may hurt another. They usually group similar tasks and assign an independent model to each group [25–27] or assign high weights on target-relevant sources [22]. The essential difference between those work and ours is that they assume a pass over the whole dataset is possible and aim to achieve the ultimate best performance, whereas we assume it is not (setting a large amount of experiment environment or maintaining a long time real data collection is costly). Consequently, they should not be considered as active.

**Passive Multi-task training/Meta learning**    While our paper focuses on data collection, some papers focus on the training process with some given dataset. For example, [22] mentioned above reweighting and joint-training all tasks. Another large topic in this scope is called "Meta-learning" [28–30], which usually focuses on more detailed updating methods. In conclusion, this line of works is parallel to our work, and all those methods can be regarded as a plug-in oracle in 1 Line 5, 9, 12.

**Sample-wise data selection for representation learning**    Classical pool-based active learning selects most informative data for a single task. Recently, some works [31–33] started to focus on selecting helpful data from a large corpus of web-scale for some known target task, where web-scale data could be seen as a mix of multi-task data without explicit "task" information. Besides, those works usually focus on coarse labels and self-contrastive learning. Therefore, although they also aim to learn a presentation/pretrained model from non-target data, their detailed settings are quite different from ours.

# B Result and analysis for target-aware

## B.1 Offline training oracles used in Algorithm

### B.1.1 Choice of $\mathcal{O}^X_{\text{offline 1}}$

To better illustrate this oracle $\mathcal{O}^X_{\text{offline}}$, we first give the following definition.

**Definition B.1** (Modified from Assumption 2 in [12]). *For any $t$ tasks with parameter matrix $\dot{V} = [\dot{v}_1, \dot{v}_2, \ldots, \dot{v}_t] \in \mathbb{R}^{d_W \times t}$. Let $\lambda_1^*$ and $\lambda_k^*$ denote the largest and smallest eigenvalues of the task diversity matrix $(k/t) B_W^{source} \dot{V} \dot{V}^\top (B_W^{source})^\top \in \mathbb{R}^{k \times k}$ respectively. Then we say $\dot{V}$ is $\mu$-incoherent, i.e.,*

$$\max_{i \in [t]} \| B_W^{source} \dot{v}_i \|^2 \leq \mu \lambda_k^*$$

Notice that here $\dot{V}$ is a general representation of collected source tasks used for training in the different stages. Therefore, the $\lambda_k^*, \mu$ is also defined differently corresponding to each stage. Specially, we have

- **Stage 1( data collected by $q_0$):**
    - $t = d_W, \dot{V} = I_{d_W}$
    - $\lambda_k^* = \frac{k}{d_W} \sigma_k^2(B_W^{\text{source}})$
    - $\mu \geq \frac{1}{\lambda_k^*}$
- **Stage 2( data collected by $q_1$):**
    - $t = k, \dot{V} = V$ where $\_, \_, V \leftarrow \text{SVD}(\hat{B}_W^{\text{source}})$ as defined in line 5
    - $\lambda_k^* = \sigma_k^2(B_W^{\text{source}})$
    - $\mu \geq \frac{\sigma_{\max}^2(B_W^{\text{source}})}{\lambda_k^*}$

Note that $\lambda_k^* = \sigma_k^2(B_W^{\text{source}})$ in the stage 2 comes from $B_W^{\text{source}} \dot{V} \dot{V}^\top (B_W^{\text{source}})^\top = \Theta(B_W^{\text{source}}(B_W^{\text{source}})^\top)$ which will be proved later. Therefore, applying these results to

Now we restate the generalization guarantees from a fixed design (passive learning)

**Theorem B.1** (Restate Theorem 1 in [12]). *Let there be $t$ linear regression tasks, each with $m$ samples, and*

$$m \geq \widetilde{\Omega}\left(\left(1 + k\left(\sigma/\sqrt{\lambda_k^*}\right)^2\right)k\log t + k^2\right), \quad \text{and } mt \geq \widetilde{\Omega}\left(\left(1 + \left(\sigma/\sqrt{\lambda_k^*}\right)^2\right)(\lambda_1^*/\lambda_k^*)\mu d_X k^2\right)$$

*Then MLLAM, initialized at $\hat{B}_X = U_{\text{init}}$ s.t. $\left\|\left(\mathbf{I} - B_X(B_X)^\top\right)U_{\text{init}}\right\|_F \leq \min\left(3/4, O\left(\sqrt{\lambda_k^*/\lambda_1^*}\right)\right)$ and run for $K = \lceil \log_2\left(\lambda_k^*\lambda_k^* mt/\lambda_1^*\sigma^2\mu d_X k^2\right)\rceil$ iterations, outputs $\hat{B}_X$ so that the following holds (w.p. $\geq 1 - K/(d_X k)^{10}$)*

$$\sin(\hat{B}_X, B_X) \leq \left\|\left(\mathbf{I} - B_X(B_X)^\top\right)\hat{B}_X\right\|_F \leq \widetilde{O}\left(\left(\frac{\sigma}{\sqrt{\lambda_k^*}}\right)k\sqrt{\frac{\mu d_X}{mt}}\right)$$

Specifically, suppose we satisfy all the requirements in the theorem and run the proper amount of times, then we can guarantee $\hat{B}_X$ after each stage $j$ with w.h.p $\geq 1 - 2K/(d_X k)^{10}$

- **Stage 1( data collected by $q_0$)**: $\sin(\hat{B}_X, B_X) \leq \widetilde{\mathcal{O}}\left(\sigma k\sqrt{\frac{d_X}{n_0}}\right)$

- **Stage 2( data collected by $q_1$)**: $\sin(\hat{B}_X, B_X) \leq \widetilde{\mathcal{O}}\left(\sigma k\sqrt{\frac{d_X \sigma_{\max}^2(B_W^{\text{source}})}{n_0}}\right)$

Let Event $\mathcal{E}_{\text{offline 1}}$ denote the above guarantees hold for all epochs.

### B.1.2 Choice of $\mathcal{O}_{\text{offline 2}}^X$

We use the ERM from [9]. For readers' convenience, we restate the formal definition of oracle below

$$\hat{B}_X = \arg\min_B \sum_{w | q_1(w) \neq 0 \text{ and } q_2(w) \neq 0} \arg\min_w \sum_{(x,y) \in Z_w} \|x^\top w - y\|_2$$

By using this ERM with the follow-up finetune on $Z_{\text{target}}$, we get the following claims. Note that this claim comes from some part of Proof of Theorem 4.1 in the previous paper and has also been used in Claim 3 in [8].

**Claim B.1.** *By running the ERM-based algorithm, we get the following upper bounds,*

$$\text{ER}(\tilde{B}_X, \nu_{target}) \leq \mathbb{E}_{w_0 \sim \nu_{target}}\left[\frac{\|P_{X_{target}\hat{B}_X}^\perp X_{target} B_X B_W w_0\|^2}{n_{target}} + \sigma^2 \frac{k + \log(1/\delta)}{n_{target}}\right]$$

We need to admit that, from a theoretical perspective, we choose this oracle since we can directly use their conclusions. But other oracles like $\mathcal{O}_{\text{offline 2}}^X$ might also work.

### B.1.3 Choice of $\mathcal{O}_{\text{offline}}^W$

This is the ERM oracle based on learned $\hat{B}_X$. Specially, we have $\hat{B}_W^{\text{source/target}} \leftarrow \mathcal{O}_{\text{offline}}^W(\{Z_w\}_{w|q(w)\neq 0}, \hat{B}_X)$ defined as

$$\hat{B}_W^{\text{source/target}} = \sum_{w|q(w)\neq 0} \hat{w}_w w^\top, \text{ where } \hat{w}_w = \arg\min_{\hat{w} \in \mathbb{R}^k} \sum_{(x,y) \sim Z_w} \|x^\top \hat{B}_X^\top \hat{w} - y\|_2,$$

## B.2 Excess risk analysis

**Theorem B.2** (Excess risk guarantees). *By running the Algo. 2, after epoch $j$, as long as $\mathcal{E}_{offline 1}$ holds, we have w.h.p $1 - \delta$,*

$$\text{ER}(\tilde{B}_X, \nu_{target}) \leq \widetilde{\mathcal{O}}\left(\sigma^2 k d_X k^* \epsilon_j^2\right)$$

*as long as*

$$\dot{n}_{target} \geq \epsilon_j^{-\frac{4}{3}} d_X^{-\frac{2}{3}} \left( k^{-\frac{2}{3}} d_W^{\frac{1}{2}} \underline{\sigma}^{-\frac{4}{3}} + k^{-\frac{4}{3}} d_W^{\frac{1}{6}} \bar{\kappa}^2 \underline{\sigma}^{-\frac{1}{3}} \right) \sqrt{k + \log(d_W/\delta)}$$

$$n_{target} \geq \epsilon_j^{-2} d_X^{-1} (k^*)^{-1} \frac{k}{k + \log(d_W/\delta)}$$

*Proof.* Here we provide the proof sketches, which will be specified in the following sections.

In Section B.2.1, we first reduce $\mathrm{ER}(\tilde{B}_X, \nu_{\mathrm{target}})$ to an optimal design problem by showing that, with a proper number of $n_{\mathrm{target}}$,

$$\mathrm{ER}(\tilde{B}_X, \nu_{\mathrm{target}}) \lesssim (kd_X + \log(1/\delta)) \mathrm{Tr}\left( \left( (B_W^{\mathrm{source}}) \left( \sum_{w \in \mathcal{S}} n_w ww^\top \right) (B_W^{\mathrm{source}})^\top \right)^{-1} B_W \left( \mathbb{E}_{\nu_{\mathrm{target}}} ww^T \right) B_W^\top \right)$$

It is easy to see that, as long as $B_W$ is known. The problem is reduced to an optimal design problem with fixed optimization target.

So the main challenge here is to iteratively estimate $B_X, B_W$ and design the budget allocation to different sources. Therefore, in Section B.2.2, we further decompose the it into

$$\mathrm{Tr}\left( \left( (B_W^{\mathrm{source}}) \left( \sum_{w \in \mathcal{S}} n_w ww^\top \right) (B_W^{\mathrm{source}})^\top \right)^{-1} B_W \left( \mathbb{E}_{\nu_{\mathrm{target}}} ww^T \right) B_W^\top \right)$$

$$\leq \underbrace{\mathbb{E}_{w_0 \sim \nu_{\mathrm{target}}} \left( \left[ (B_W w_0)^\top \square B_W w_0 \right] - \mathrm{Tr}\left( \beta_3 (B_W W')^\top \square B_W W' \right) \right)}_{\text{target agnostic exploration error}} + \beta_3 \underbrace{\mathrm{Tr}\left( (B_W W')^\top \square B_W W' \right)}_{\text{target-aware exploration error}}$$

where $\square = \left( B_W \left( \sum_{w \in \mathcal{S}} n_w ww^\top \right) B_W^\top \right)^{-1}$. Here the **target-aware exploration error** captures the error from selecting the target-related sources (defined by $q_2$). On the other hand, the **target agnostic exploration error** captures the error from model estimation and the uniform exploration.

Now the main challenge here is to upper-bound the model estimation error. Specifically, the estimation comes from Coarse exploration (Stage 1) and Fine target-agnostic exploration (Stage 2). Specifically, in Section B.2.3, we show that the $k$-dim-subspace represented by $q_1$ is a good course approximation up to *multiplicative* error. Then in Section B.2.4, we further tight the upper bound using data collected according to up to some *additive* error. □

### B.2.1 Reduce to an optimal design problem

For any fixed epoch $j$, let $n_w^j$ denotes the samples collected so far for task $w$ and $\mathcal{S}$ denotes the set of tasks used in computing $\tilde{B}_X$. Therefore, we have $\mathcal{S} = \{w | q_1(w) \neq 0 \text{ and } \tilde{q}(w) \neq 0\}$ and $n_w \geq n_2(w) + n_2^j(w)$. For convenience, we omit the superscript $j$ in the rest of the proofs.

From Claim B.1, it is easy to see that our main target is to optimize $\mathbb{E}_{w_0 \sim \nu_{\mathrm{target}}} \| P_{X_{\mathrm{target}} \hat{B}_X}^\perp X_{\mathrm{target}} B_X B_W w_0 \|^2$. Decompose $B_W \left( \sum_{w \in \mathcal{S}} n_w ww^\top \right) B_W^\top$ as $UDU^\top$ and let $\Sigma_W = U\sqrt{D}U^\top$. As long as $\Sigma_W$ is full rank, which we will prove later in Section B.2.3, we

have with probability $1 - \delta$,

$$\mathbb{E}_{w_0 \sim \nu_{\text{target}}} \| P^{\perp}_{X_{\text{target}} \hat{B}_X} X_{\text{target}} B_X B_W w_0 \|^2$$

$$= \mathbb{E}_{w_0 \sim \nu_{\text{target}}} \| P^{\perp}_{X_{\text{target}} \hat{B}_X} X_{\text{target}} B_X \Sigma_W^{\frac{1}{2}} \Sigma_W^{-\frac{1}{2}} B_W w_0 \|^2$$

$$\leq \mathbb{E}_{w_0 \sim \nu_{\text{target}}} \| P^{\perp}_{X_{\text{target}} \hat{B}_X} X_{\text{target}} B_X \Sigma_W^{\frac{1}{2}} \|_F^2 \| \Sigma_W^{-\frac{1}{2}} B_W w_0 \|^2$$

$$= \mathbb{E}_{w_0 \sim \nu_{\text{target}}} \| P^{\perp}_{X_{\text{target}} \hat{B}_X} X_{\text{target}} B_X B_W \tilde{W}_{\mathcal{S}} \|_F^2 (B_W w_0)^{\top} \left( B_W \left( \sum_{w \in \mathcal{S}} n_w w w^{\top} \right) B_W^{\top} \right)^{-1} B_W w_0$$

$$= \| P^{\perp}_{X_{\text{target}} \hat{B}_X} X_{\text{target}} B_X B_W \tilde{W}_{\mathcal{S}} \|_F^2 \mathrm{Tr} \left( \left( B_W \left( \sum_{w \in \mathcal{S}} n_w w w^{\top} \right) B_W^{\top} \right)^{-1} B_W \left( \mathbb{E}_{\nu_{\text{target}} \in \Delta(\mathcal{W}_{\text{target}})} w w^T \right) B_W^{\top} \right)$$

$$\lesssim \sigma^2 n_{\text{target}} \left( k d_X + \log(1/\delta) \right) \mathrm{Tr} \left( \left( B_W \left( \sum_{w \in \mathcal{S}} n_w w w^{\top} \right) B_W^{\top} \right)^{-1} B_W \left( \mathbb{E}_{\nu_{\text{target}}} w w^T \right) B_W^{\top} \right)$$

$$= \sigma^2 n_{\text{target}} \left( k d_X + \log(1/\delta) \right) \mathrm{Tr} \left( \left( (B_W^{\text{source}}) \left( \sum_{w \in \mathcal{S}} n_w w w^{\top} \right) (B_W^{\text{source}})^{\top} \right)^{-1} B_W \left( \mathbb{E}_{\nu_{\text{target}}} w w^T \right) B_W^{\top} \right)$$

Therefore, we aim to minimize the $\mathrm{Tr} \left( \left( (B_W^{\text{source}}) \left( \sum_{w \in \mathcal{S}} n_w w w^{\top} \right) (B_W^{\text{source}})^{\top} \right)^{-1} B_W \left( \mathbb{E}_{\nu_{\text{target}}} w w^T \right) B_W^{\top} \right)$.
As we mentioned before, this is a pure optimal design problem if $B_W$ is known in advance.

### B.2.2 Bound decomposition and the excess risk result

Let $\square = \left( B_W \left( \sum_{w \in \mathcal{S}} n_w w w^{\top} \right) B_W^{\top} \right)^{-1}$, we have

$$\mathbb{E}_{w_0 \sim \nu_{\text{target}}} \left[ (B_W w_0)^{\top} \square B_W w_0 \right]$$
$$= \underbrace{\mathbb{E}_{w_0 \sim \nu_{\text{target}}} \left( \left[ (B_W w_0)^{\top} \square B_W w_0 \right] - \mathrm{Tr} \left( \beta_3 (B_W W')^{\top} \square B_W W' \right) \right.}_{\text{target agnostic exploration error}} + \beta_3 \underbrace{\mathrm{Tr} \left( (B_W W')^{\top} \square B_W W' \right)}_{\text{target-aware exploration error}}$$

**We first deal with the target-aware exploration error**. It is easy to see that

$$\beta_3 \mathrm{Tr} \left( (B_W W')^{\top} \square B_W W' \right)$$

$$= \beta_3 \mathrm{Tr} \left( \left( B_W \sum_w q_1(w) n_1 w w^{\top} (B_W)^{\top} + B_W \sum_w q_2(w) n_2 w w^{\top} (B_W)^{\top} \right)^{-1} B_W W' (B_W W')^{\top} \right)$$

$$\leq \mathrm{Tr} \left( \left( \max_i \| \tilde{W}(i) \|_{2(\infty)}^2 B_W \epsilon^{-2} \tilde{W} \tilde{W}^{\top} (B_W)^{\top} \right)^{-1} B_W W' (B_W W')^{\top} \right)$$

$$\leq \mathrm{Tr} \left( \left( B_W \epsilon^{-2} W' (W')^{\top} (B_W)^{\top} \right)^{-1} B_W W' (B_W W')^{\top} \right)$$

$$= \epsilon^2 \mathrm{rank}(\hat{B}_W W' (W')^{\top} \hat{B}_W^{\top})$$

$$\leq \epsilon^2 \mathrm{rank}(B_W \mathbb{E}_{\nu_{\text{target}}} [w_0 w_0^{\top}] B_W^{\top})$$

where the last equality comes from Lemma B.5.

**We then deal with the target-agnostic exploration term.** Let the clipping threshold in Line 9 be $\bar{\gamma}_j$. That is, ignoring all $\lambda_i \leq \bar{\gamma}$. Now, for $\beta_3 \geq 8$, when event $\mathcal{E}_{\text{offline 1}}$, holds, we have w.h.p $1 - d_W \delta$

$$\mathbb{E}_{w_0 \sim \nu_0} \left[ (B_W w_0)^\top \square B_W w_0 \right] - \beta_3 \text{Tr} \left( (B_W W')^\top \square B_W W' \right)$$

$$= \mathbb{E}_{w_0 \sim \nu_0} \text{Tr} \left( \square \left( B_W w_0 (B_W w_0)^\top - 4 \hat{B}_W^{\text{target}} w_0 (\hat{B}_W^{\text{target}} w_0)^\top \right) \right)$$

$$\quad + \mathbb{E}_{w_0 \sim \nu_0} \text{Tr} \left( \square \left( \frac{1}{2} \beta_3 \hat{B}_W^{\text{source}} w' (\hat{B}_W w')^\top - \beta_3 B_W^{\text{source}} w' (B_W w')^\top \right) \right)$$

$$\quad + \mathbb{E}_{w_0 \sim \nu_0} \text{Tr} \left( \square \left( 4 \hat{B}_W^{\text{target}} w_0 (\hat{B}_W^{\text{target}} w_0)^\top - \frac{1}{2} \beta_3 \hat{B}_W^{\text{source}} W' (\hat{B}_W^{\text{source}} W')^\top \right) \right)$$

$$\leq \mathbb{E}_{w_0 \sim \nu_0} \text{Tr} \left( \left( 4 B_W^{\text{target}} w_0 (B_W^{\text{target}} w_0)^\top - 4 \hat{B}_W^{\text{target}} w_0 (\hat{B}_W^{\text{target}} w_0)^\top \right) \right) \| \square \|$$

$$\quad + \beta_3 \text{Tr} \left( \left( \frac{1}{2} \hat{B}_W^{\text{source}} w' (\hat{B}_W w')^\top - \frac{1}{2} \dot{B}_W W' (\dot{B}_W W')^\top \right) \right) \| \square \|$$

$$\quad + k \bar{\gamma} \| \square \|$$

$$\leq \| \square \| \| \mathbb{E}[w_0 w_0^T] \|_* \| (B_W^{\text{target}})^\top B_W^{\text{target}} - (\hat{B}_W^{\text{target}})^\top \hat{B}_W^{\text{target}} \| + \| \square \| \| W' (W')^\top \|_* \| \dot{B}_W^\top \dot{B}_W - \hat{B}_W^\top \hat{B}_W \| + k \bar{\gamma} \| \square \|$$

$$\leq 2 \| \square \| \| \mathbb{E}[w_0 w_0^T] \|_* \| B_W^{\text{target}} - \hat{B}_W^{\text{target}} \| \| B_W^{\text{target}} \|$$

$$\quad + 2 \| \square \| \| W' (W')^\top \|_* \| \dot{B}_W - \hat{B}_W^{\text{source}} \| \| \hat{B}_W^{\text{source}} \|$$

$$\quad + k \bar{\gamma} \| \square \|$$

$$\leq \epsilon^2$$

where the second two terms in the first inequality come from Section B.2.3 and the last term in the first inequality comes from the definition of $W'$. Here $\dot{B}_W = B_W V V^\top = B_W^{\text{source}} V V^\top$ is a pseudo representation of $B_W^{\text{source}}$, where $V$ is the one calculated in Line 5. And the last inequality comes from the results in Section B.2.4. Notice that the probability $1 - d_W \delta$ comes from the union bound on all the calls of $\mathcal{O}_{\text{offline}}^W$.

**Now combine the bounds above, we have**

$$\text{ER}(\tilde{B}_X, \nu_{\text{target}}) \leq \sigma^2 \left( k d_X \log((\kappa N_i)/d_W) + \log \frac{1}{\delta} \right) k^* \epsilon^2$$

### B.2.3 Detail proofs for warm-up stage

After the first stage, according to Section B.1.1, as long as $\mathcal{E}_{\text{offline 1}}$ holds, we have

$$\sin(\hat{B}_X, B_X) \leq \widetilde{\mathcal{O}} \left( \sigma k \sqrt{\frac{d_X}{n_0}} \right)$$

Therefore, by Lemma B.2, we have with probability $1 - d_W \delta$,

$$\| \hat{B}_W^{\text{source}} - B_W^{\text{source}} \| \leq 2 \sqrt{k} \sin(\hat{B}_X, B_X) \| B_W \| + \sqrt{\frac{d_W}{n_0}} (k + \log(2/\delta)^{\frac{1}{4}} d_W^{\frac{1}{4}}$$

$$\leq 2 k^{\frac{3}{2}} \sqrt{\frac{d_X}{n_0}} \| B_W \| + 2 d_W^{\frac{3}{4}} (k + \log(2/\delta)^{\frac{1}{4}} \sqrt{\frac{1}{n_0}}$$

As long as $n_0 \geq 1024 \bar{\kappa}^2 \left( k^3 d_X \bar{\kappa}^2 + \frac{d_W^{\frac{3}{2}}}{\underline{\sigma}^2} \sqrt{k + \log(1/\delta)} \right)$, by using the Lemma B.1 below, we have for any arbitrary matrix $M$,

$$\frac{1}{2} B_W M (B_W)^\top \leq \dot{B}_W M \dot{B}_W^\top \leq \frac{3}{2} B_W M (B_W)^\top$$

In the other word, $\dot{B}$ can be regarded as a pseudo representation of $B_W^{\text{source}}$. In all the later epochs, when exploring $k$-subspace according to $q_1^j$, the learner actually learns $\dot{B}_W$.

**Lemma B.1** (Guarantee on exploration basis 1). *Suppose we have the estimated $\hat{B}_W$ satisfies*

$$8\|B_W - \hat{B}_W\|\|B_W\| \leq \frac{1}{2}\lambda_{\min}(B_W B_W^\top)$$

$$\dot{V} \leftarrow \text{column space of } \mathrm{SVD}(\hat{B}_W),$$

*then let $\dot{B}_W = B_W \dot{V}\dot{V}^\top$, we have, for any arbitrary matrix $M$,*

$$\frac{1}{2}B_W M (B_W)^\top \leq \dot{B}_W M \dot{B}_W^\top \leq \frac{3}{2}B_W M (B_W)^\top$$

*Proof.*

$$\dot{B}_W M \dot{B}_W^\top - B_W M (B_W)^\top$$
$$= \dot{B}_W M \dot{B}_W^\top - \hat{B}_W M (\hat{B}_W)^\top + \hat{B}_W M (\hat{B}_W)^\top - B_W M B_W^\top$$
$$= (\dot{B}_W - \hat{B}_W)M B_W^\top + \hat{B}_W M (\dot{B}_W - \hat{B}_W)^\top + (\hat{B}_W - B_W)M(\hat{B}_W)^\top + B_W M(\hat{B}_W - B_W)^\top$$
$$= (B_W - \hat{B}_W)\dot{V}\dot{V}^\top M B_W^\top + \hat{B}_W M \dot{V}\dot{V}^\top(B_W - \hat{B}_W)^\top + (\hat{B}_W - B_W)M(\hat{B}_W)^\top + B_W M(\hat{B}_W - B_W)^\top$$

Therefore, according to our assumption, we can upper bound the above as

$$\dot{B}_W M \dot{B}_W^\top - B_W M (B_W)^\top \leq 2\|B_W - \hat{B}_W\|\left(\|\hat{B}_W\| + \|B_W\|\right)M$$
$$\leq \left(4\|B_W - \hat{B}_W\|\|B_W\| + 2\|B_W - \hat{B}_W\|_2^2\right)M$$
$$\leq 8\|B_W - \hat{B}_W\|\|B_W\|M$$
$$\leq \frac{1}{2}\lambda_{\min}(B_W B_W^\top)M \leq \frac{1}{2}B_W M B_W^\top$$

Similarly, it can be lower bounded by $-\frac{1}{2}B_W M B_W^\top$. Therefore we can get the target result by rearranging. $\square$

### B.2.4 Detail proofs for task-agnostic exploration strategy

First, we upper bound two $\|B_W - \hat{B}_W\|$ terms. From section B.1.1, as long as $\mathcal{E}_{\text{offline 1}}$ holds, we have

$$\sin(\hat{B}_X, B_X) \leq \widetilde{\mathcal{O}}\left(k\sqrt{\frac{d_X}{n_1}}\|B_W^{\text{source}}\|\right)$$

Therefore, by Lemma B.2, we have w.h.p at least $1 - (k + d_W^{\text{target}})\delta$

$$\|\hat{B}_W^{\text{source}} - B_W^{\text{source}}\| \leq 2\sqrt{k}\sin(\hat{B}_X, B_X)\|B_W^{\text{source}}\| + \sqrt{\frac{k}{n_1}}(k + \log(2/\delta)^{\frac{1}{4}}k^{\frac{1}{4}}$$
$$\leq 2k^{\frac{3}{2}}\sqrt{\frac{d_X}{n_1}}\|B_W^{\text{source}}\|^2$$

$$\|\hat{B}_W^{\text{target}} - B_W^{\text{target}}\| \leq 2k\sin(\hat{B}_X, B_X)\|B_W^{\text{target}}\| + \sqrt{\frac{1}{\dot{n}_{\text{target}}}}(k + \log(2/\delta)^{\frac{1}{4}}(d_W^{\text{target}})^{\frac{1}{4}}$$
$$\leq 2k^{\frac{3}{2}}\sqrt{\frac{d_X}{n_1}}\|B_W^{\text{target}}\|^2 + 2\sqrt{\frac{1}{\dot{n}_{\text{target}}}}(k + \log(2/\delta)^{\frac{1}{4}}(d_W^{\text{target}})^{\frac{1}{4}}$$
$$\leq 4k^{\frac{3}{2}}\sqrt{\frac{d_X}{n_1}}\|B_W^{\text{target}}\|^2$$

where the last equality holds as long as $\dot{n}_{\text{target}} \geq n_1\frac{\sqrt{(k+\log(2/\delta))d_W^{\text{target}}}}{k^3 d_X\|B_W^{\text{target}}\|^2}$.

Next, we upper bound the $\|W'(W')\|$ according to Lemma B.7.

$$\|W'(W')^\top\|_* \lesssim \frac{1}{\sigma_{\min}^2(B_W^{\text{source}})}\|B_W^{\text{target}}\mathbb{E}_{\nu_{\text{target}}}[w_0 w_0^\top](B_W^{\text{target}})^\top\|_*$$

$$\leq \frac{1}{\sigma_{\min}^2(B_W^{\text{source}})}\|B_W^{\text{target}}\|^2\|\mathbb{E}_{\nu_{\text{target}}}[w_0 w_0^\top]\|_*$$

$$\leq \frac{1}{\sigma_{\min}^2(B_W^{\text{source}})}\|B_W^{\text{target}}\|^2$$

Finally, we have, by definition

$$\bar{\gamma} \leq 2\|\hat{B}_W^{\text{target}} - B_W^{\text{target}}\|\|B_W^{\text{target}}\|$$

$$\|\square\| \leq \frac{k}{n_1 \sigma_{\min}^2(\dot{B}_W)} \lesssim \frac{k}{n_1 \sigma_{\min}^2(B_W^{\text{source}})}$$

Combine all above, we have the upper bound

$$\|\square\| \left( \|\mathbb{E}[w_0 w_0^T]\|_* \|B_W^{\text{target}} - \hat{B}_W^{\text{target}}\|\|B_W^{\text{target}}\| + \|\|W'(W')^\top\|_*\|\dot{B}_W - \hat{B}_W^{\text{source}}\|\|B_W^{\text{source}}\| + k\bar{\gamma} \right)$$

$$\lesssim \frac{k}{n_1 \sigma_{\min}^2(B_W^{\text{source}})} * k^{\frac{3}{2}}\sqrt{\frac{d_X}{n_1}} * \left( k\|B_W^{\text{target}}\|^3 + \|B_W^{\text{source}}\|\|B_W^{\text{target}}\|^2\bar{\kappa}^2 \right)$$

$$\leq k^{\frac{5}{2}}d_X^{\frac{1}{2}}n_1^{-\frac{3}{2}}\|B_W^{\text{target}}\|^2 \left( \frac{k\|B_W^{\text{target}}\|}{\underline{\sigma}^2} + \frac{\bar{\kappa}^3}{\underline{\sigma}} \right)$$

$$\leq k^{\frac{5}{2}}d_X^{\frac{1}{2}}n_1^{-\frac{3}{2}}d_W \left( k\sqrt{d_W}\underline{\sigma}^{-2} + \bar{\kappa}^3\underline{\sigma}^{-1} \right)$$

As long as $n_1 \geq \epsilon_j^{-\frac{4}{3}}k^{\frac{5}{3}}d_W^{\frac{2}{3}}d_X^{\frac{1}{3}} \left( k^{\frac{2}{3}}d_W^{\frac{1}{3}}\underline{\sigma}^{-\frac{4}{3}} + \bar{\kappa}^2\underline{\sigma}^{-\frac{2}{3}} \right)$, we have the final bound $\epsilon_j^2$.

### B.2.5 Auxillary lemmas

**Lemma B.2.** *Consider any $t$ regression tasks parameterized by $\{\dot{v}_i\}_{i\in[n]}$. Denote $\dot{V} = [\dot{v}_1, \dot{v}_2, \ldots, \dot{v}_t]$ and $|X_{\dot{v}_i}| = n$ for all $i \in [t]$, define*

$$\hat{B}_W = \sum_{i\in k}\hat{w}_i\dot{v}_i^\top, \text{ where } \hat{w}_i = \underset{w\in\mathbb{R}^k}{\arg\min}\|X_{\dot{v}_i}\hat{B}_X^\top w - Y_{\dot{v}_i}\|_2,$$

*then we have with probability at least $1 - \delta$,*

$$\|\hat{B}_W - \dot{B}_W\| = \|\hat{B}_W - B_W\dot{V}\dot{V}^\top\| \leq 2\sqrt{k}\sin(\hat{B}_X, B_X)\|\dot{B}_W\| + \sqrt{\frac{1}{n}}(k + \log(2/\delta)^{\frac{1}{4}}|\dot{V}|^{\frac{1}{4}}$$

*Proof.* From [8], we get that the explicit form of $\hat{w}_i$, which is the estimation of actual $B_w\dot{v}_i$ as

$$\left( \hat{B}_X X_{\dot{v}_i}^\top X_{\dot{v}_i}\hat{B}_X^\top \right)^{-1}\hat{B}_X X_{\dot{v}_i}^\top X_{\dot{v}_i}B_X^\top B_X\dot{v}_i + \left( \hat{B}_X X_{\dot{v}_i}^\top X_{\dot{v}_i}\hat{B}_X^\top \right)^{-1}\hat{B}_X X_{\dot{v}_i}^\top\xi_w$$

By abusing notation a little bit, here we use subscription $i$ to denote the items that associate the task encoded by $\dot{v}_i$. Therefore, we have

$$
\begin{aligned}
\hat{B}_W &= \sum_{i=1}^{t} \left( \hat{B}_X X_i^\top X_i \hat{B}_X^\top \right)^{-1} \hat{B}_X X_i^\top X_i B_X^\top B_W \dot{v}_i \dot{v}_i^\top + \sum_{i=1}^{t} \left( \hat{B}_X X_i^\top X_i \hat{B}_X^\top \right)^{-1} \hat{B}_X X_i^\top \xi_i \dot{v}_i^\top \\
&= \sum_{i=1}^{t} \left( \hat{B}_X X_i^\top X_i \hat{B}_X^\top \right)^{-1} \hat{B}_X X_i^\top X_i \left( \hat{B}_X^\top \hat{B}_X + \hat{B}_{X,\perp}^\top \hat{B}_{X,\perp} \right) B_X^\top B_W \dot{v}_i \dot{v}_i^\top \\
&\quad + \sum_{i=1}^{t} \left( \hat{B}_X X_i^\top X_i \hat{B}_X^\top \right)^{-1} \hat{B}_X X_i^\top \xi_i \dot{v}_i^\top \\
&= \hat{B}_X B_X^\top \dot{B}_W + \sum_{i=1}^{t} \left( \hat{B}_X X_i^\top X_i \hat{B}_X^\top \right)^{-1} \hat{B}_X X_i^\top X_i \hat{B}_{X,\perp}^\top \hat{B}_{X,\perp} B_X^\top B_W \dot{v}_i \dot{v}_i^\top \\
&\quad + \sum_{i=1}^{d} \left( \hat{B}_X X_i^\top X_i \hat{B}_X^\top \right)^{-1} \hat{B}_X X_i^\top \xi_i \dot{v}_i^\top
\end{aligned}
$$

And the estimation difference between $B_W, \hat{B}_W$ can be decomposed into three parts

$$
\begin{aligned}
\| \dot{B}_W - \hat{B}_W \| &\leq \| \left( \hat{B}_X B_X^\top - I_k \right) \dot{B}_W \| \\
&\quad + \| \sum_{i=1}^{t} \left( \hat{B}_X X_i^\top X_i \hat{B}_X^\top \right)^{-1} \hat{B}_X X_i^\top X_i \hat{B}_{X,\perp}^\top \hat{B}_{X,\perp} B_X^\top B_W \dot{v}_i \dot{v}_i^\top \| \\
&\quad + \| \sum_{i=1}^{t} \left( \hat{B}_X X_i^\top X_i \hat{B}_X^\top \right)^{-1} \hat{B}_X X_i^\top \xi_i \dot{v}_i^\top \| \\
&\leq \| \left( \hat{B}_X B_X^\top - I_k \right) \| \| \dot{B}_W \| \\
&\quad + \max_i \| \left( \hat{B}_X X_i^\top X_i \hat{B}_X^\top \right)^{-1} \hat{B}_X X_i^\top X_i \hat{B}_{X,\perp}^\top \hat{B}_{X,\perp} B_X^\top \| \| \sum_{i=1}^{t} B_W \dot{v}_i \dot{v}_i^\top \| \\
&\quad + \| \sum_{i=1}^{t} \left( \hat{B}_X X_i^\top X_i \hat{B}_X^\top \right)^{-1} \hat{B}_X X_i^\top \xi_i \dot{v}_i^\top \|
\end{aligned}
$$

By using Lemma B.3 and Lemma B.4, we can bound the first two terms by

$$
2\sqrt{k} \sin(\hat{B}_X, B_X) \| \dot{B}_W \|
$$

Now we are going to bound the last term which is the noise term.

$$
\begin{aligned}
&\| \sum_{i=1}^{|\dot{V}|} \left( \hat{B}_X X_i^\top X_i \hat{B}_X^\top \right)^{-1} \hat{B}_X X_i^\top \xi_i \dot{v}_i^\top \|^2 \\
&= \lambda_{\max} \left( \sum_{i=1}^{|\dot{V}|} \left( \hat{B}_X X_i^\top X_i \hat{B}_X^\top \right)^{-1} \hat{B}_X X_i^\top \xi_i \dot{v}_i^\top \right) \left( \sum_{i=1}^{|\dot{V}|} \left( \hat{B}_X X_i^\top X_i \hat{B}_X^\top \right)^{-1} \hat{B}_X X_i^\top \xi_i \dot{v}_i^\top \right)^\top \\
&\leq \lambda_{\max} \left( \sum_{i=1}^{|\dot{V}|} \left( \hat{B}_X X_i^\top X_i \hat{B}_X^\top \right)^{-1} \hat{B}_X X_i^\top \xi_i \xi_i^\top X_i \hat{B}_X^\top \left( \hat{B}_X X_i^\top X_i \hat{B}_X^\top \right)^{-1} \right)
\end{aligned}
$$

Note that, $x_i \sim \mathcal{N}(0, I_d)$ and

$$
\begin{aligned}
\left( \hat{B}_X X_i^\top X_i \hat{B}_X^\top \right)^{-1} \hat{B}_X X_i^\top \xi_i &\sim \mathcal{N} \left( 0, \left( \left( \hat{B}_X X_i^\top X_i \hat{B}_X^\top \right)^{-1} \hat{B}_X X_i^\top X_i \hat{B}_X^\top \left( \hat{B}_X X_i^\top X_i \hat{B}_X^\top \right)^{-1} \right) \right) \\
&\sim \mathcal{N} \left( 0, \left( \hat{B}_X X_i^\top X_i \hat{B}_X^\top \right)^{-1} \right)
\end{aligned}
$$

Therefore, by the concentration inequality of the covariance matrix, we have, w.h.p $1 - \delta$,

$$\lambda_{\max} \left( \sum_{i=1}^{|\dot{V}|} \left( \hat{B}_X X_i^\top X_i \hat{B}_X^\top \right)^{-1} \hat{B}_X X_i^\top \xi_i \xi_i^\top X_i \hat{B}_X^\top \left( \hat{B}_X X_i^\top X_i \hat{B}_X^\top \right)^{-1} \right) \leq \frac{1}{n} \sqrt{(k + \log(2/\delta)|\dot{V}|}$$

Combining everything above, we have the final bound. $\square$

**Lemma B.3.** *Given $\hat{B}_X, B_X$ are orthonormal matrices, as well as $\mathbb{E}[xx^T] = I_{d_X}$ for all tasks $w$, we have*

$$\|I_k - \hat{B}_X B_X^\top\| \leq \mathcal{O}\left( \sqrt{k} \sin(\hat{B}_X, \hat{B}_X) \right)$$

*Proof.* Denote $B_X \hat{B}_X^\top = UDV^\top$, by definition, we have $D = \text{diag}(\cos\theta_1, \cos\theta_2, \ldots, \cos\theta_k)$ from the largest singular value to minimum singular value and $\sin\theta_k \leq \sin(\hat{B}_X, \hat{B}_X)$. Therefore we have,

$$\text{Tr}(\hat{B}_X B_X^\top) \geq k\sqrt{1 - \sin^2(\hat{B}_X, B_X)} \geq k - k\sin^2(\hat{B}_X, B_X)$$

And

$$\|I_k - \hat{B}_X B_X^\top\|^2 = \lambda_{\max} \left( I_k - \hat{B}_X B_X^\top \right)^\top \left( I_k - \hat{B}_X B_X^\top \right)$$

$$\leq \text{Tr} \left( I_k - \hat{B}_X B_X^\top \right)^\top \left( I_k - \hat{B}_X B_X^\top \right)$$

$$\leq \text{Tr} \left( I_k + \left( \hat{B}_X B_X^\top \right)^\top \hat{B}_X B_X^\top - \left( \hat{B}_X B_X^\top \right)^\top - \hat{B}_X B_X^\top \right)$$

$$\leq 2k - 2k + 2k\sin^2(\hat{B}_X, B_X) \leq 2k\sin^2(\hat{B}_X, B_X)$$

$\square$

**Lemma B.4** (Restate from [11]). *Given $\hat{B}_X, B_X$ are orthonormal matrices, as well as $\mathbb{E}[xx^T] = I_{d_X}$ for any fixed task $w$, we have*

$$\| \left( \hat{B}_X X_w^\top X_w \hat{B}_X^\top \right)^{-1} \hat{B}_X X_w^\top X_w \hat{B}_{X,\perp}^\top \hat{B}_{X,\perp} B_X^\top \| \leq \sin(\hat{B}_X, \hat{B}_X)$$

*Proof.* Here we follow the same proof step as in [11]. (Bound on the second error term in Lemma 19)

$$\| \left( \hat{B}_X X_w^\top X_w \hat{B}_X^\top \right)^{-1} \hat{B}_X X_w^\top X_w \hat{B}_{X,\perp}^\top \hat{B}_{X,\perp} B_X^\top \|$$

$$\leq \| \left( \hat{B}_X X_w^\top X_w \hat{B}_X^\top \right)^{-1} \hat{B}_X X_w^\top X_w \hat{B}_{X,\perp}^\top \| \sin(\hat{B}_X, \hat{B}_X)$$

$$\leq \sin(\hat{B}_X, \hat{B}_X)$$

$\square$

## B.3 Lemmas about the properties of $W'$

**Lemma B.5.**
$$rank(\hat{B}_W W W' \hat{B}_W^\top) \leq rank(B_W \mathbb{E}_{\nu_{target}}[w_0 w_0^\top] B_W^\top)$$

*Proof.* By using Welys inequality, we have for any eigenvalue $i \in [k]$,

$$|\lambda_i \left( \hat{B}_W^{\text{target}} \mathbb{E}[w_0 w_0^\top](\hat{B}_W^{\text{target}})^\top \right) - \lambda_i \left( B_W^{\text{target}} \mathbb{E}[w_0 w_0^\top](B_W^{\text{target}})^\top \right)|$$

$$\leq \|\hat{B}_W^{\text{target}} \mathbb{E}[w_0 w_0^\top](\hat{B}_W^{\text{target}})^\top - B_W^{\text{target}} \mathbb{E}[w_0 w_0^\top](B_W^{\text{target}})^\top\|$$

$$\leq \|\hat{B}_W^{\text{target}} (\hat{B}_W^{\text{target}})^\top - B_W^{\text{target}} (B_W^{\text{target}})^\top\|$$

$$\leq 2\|\hat{B}_W^{\text{target}} - B_W^{\text{target}}\| \|B_W^{\text{target}}\|$$

$$\leq \left( 2k^{\frac{3}{2}} \sqrt{\frac{d_X}{n_1}} \|B_W^{\text{source}}\|^2 + 2\sqrt{\frac{k}{\dot{n}_{\text{target}}}} \right) \|B_W^{\text{target}}\|$$

where the last inequality comes from Lemma B.2 and the fact $\sin(\hat{B}_X, B_X) \leq \widetilde{\mathcal{O}}\left(k\sqrt{\frac{d_X}{n_1}}\|B_W^{\text{source}}\|\right)$. Therefore, for all the $i \geq k^*$,

$$\lambda_i\left(\hat{B}_W^{\text{target}}\mathbb{E}[w_0 w_0^\top](\hat{B}_W^{\text{target}})^\top\right) \geq \left(2k\sqrt{\frac{d_X}{n_1}}\|B_W^{\text{source}}\|^2 + 2\sqrt{\frac{k}{\dot{n}_{\text{target}}}}\right)\|B_W^{\text{target}}\|$$

Clipping those non-significant directions leads to the result. $\qquad\square$

**Lemma B.6.** *Define* $W_i^* = \arg\min_v \|v\|_2 \quad, s.t. \hat{B}_W^{source} v = \hat{u}_i \hat{\Lambda}_i$, *we have*
$$\max_i \|W_i'\| \leq \min\{k^*, \kappa^2(B_W^{source})\} \max_i \|W_i^*\|$$

*Proof.* By definition of $W'$, we have, for any $W_i'$,
$$W_i' = \arg\min_v \|v\|_2 \quad, s.t. \hat{B}_W^{\text{source}} v = \hat{u}_i \hat{\Lambda}_i$$

where $\hat{U}, \hat{\Lambda}_i \leftarrow \text{Eig}(\mathbb{E}[\hat{B}_W^{\text{target}} w_0 w_0^\top (\hat{B}_W^{\text{target}})^\top])$. By solving this optimization, we get
$$W_i' = (\hat{B}_W^{\text{source}})^\top \left(\hat{B}_W^{\text{source}}(\hat{B}_W^{\text{source}})^\top\right)^{-1} \hat{u}_i\sqrt{\hat{\Lambda}_i}$$

and therefore,

$$\max_i \|W_i'\|^2 = \max_i \sqrt{\hat{\lambda}_i}\hat{u}_i^\top \left(\hat{B}_W^{\text{source}}(\hat{B}_W^{\text{source}})^\top\right)^{-1}(\hat{B}_W^{\text{source}})(\hat{B}_W^{\text{source}})^\top \left(\hat{B}_W^{\text{source}}(\hat{B}_W^{\text{source}})^\top\right)^{-1}\hat{u}_i\hat{\Lambda}_i$$

$$= \max_i \sqrt{\hat{\lambda}_i}\hat{u}_i^\top \left(\hat{B}_W^{\text{source}}(\hat{B}_W^{\text{source}})^\top\right)^{-1}\hat{u}_i\sqrt{\hat{\Lambda}_i}$$

$$\lesssim \max_i \hat{\lambda}_i\hat{u}_i^\top \left(B_W^{\text{source}}(B_W^{\text{source}})^\top\right)^{-1}\hat{u}_i$$

where the last inequality comes from Lemma B.1. Similarly, the ground truth $W^*$ can be represented as

$$\max_i \|W_i^*\|^2 = \max_i \lambda_i u_i^\top \left(B_W^{\text{source}}(B_W^{\text{source}})^\top\right)^{-1} u_i$$
$$\text{where, } \mathbb{E}_{w_0}\left[B_W^{\text{target}} w_0 w_0^\top (B_W^{\text{target}})^\top\right].$$

and denote $H = \hat{U}\hat{\Lambda}\hat{U}^\top - \mathbb{E}_{w_0}\left[B_W^{\text{target}} w_0 w_0^\top (B_W^{\text{target}})^\top\right]$.

Now we are now going to upper bound $\max_i \|W_i'\|$ in terms of $\max_i \|W_i^*\|$. Suppose $j = \arg\max \|W_i'\|$ and $B_W^{\text{target}} = U\Lambda U^\top$.

Firstly, we will lower bound the $\hat{\lambda}_i$. Given $\|\mathbb{E}_{w_0}\left[B_W^{\text{target}} w_0 w_0^\top (B_W^{\text{target}})^\top\right]\| \leq \frac{1}{2k}$, we can always found an $\|W_i'\|^2 \geq \frac{1}{2k\sigma_{\max}^2(B_W^{\text{source}})}$. Therefore, we have

$$\hat{\lambda}_j \geq \frac{1}{2k\kappa(B_W^{\text{source}})}$$

Then we consider the following two cases.

**(Case 1) When $\kappa(B_W^{\text{source}})$ is small:** By Wely's inequality, there always exists some $u_m, \lambda_m$ that $\hat{\lambda}_j \leq \mathcal{O}(\lambda_m)$. Therefore,

$$\hat{\lambda}_j\hat{u}_j^\top \left(B_W^{\text{source}}(B_W^{\text{source}})^\top\right)^{-1}\hat{u}_j \leq \hat{\lambda}_j u_m^\top \left(B_W^{\text{source}}(B_W^{\text{source}})^\top\right)^{-1} u_m\kappa(B_W^{\text{source}})^2$$

$$\leq \lambda_m u_m^\top \left(B_W^{\text{source}}(B_W^{\text{source}})^\top\right)^{-1} u_m\kappa(B_W^{\text{source}})^2$$

$$\leq \max_i \|W_i^*\|^2\kappa(B_W^{\text{source}})^2$$

**(Case 2) When $\kappa(B_W^{\text{source}})$ is large:** Decompose $\hat{B}_W^{\text{target}} W'(W')^\top (\hat{B}_W^{\text{target}})^\top$ as follows

$$\hat{B}_W^{\text{target}} W'(W')^\top (\hat{B}_W^{\text{target}})^\top = \hat{U}_0\hat{\Lambda}_0\hat{U}_0^\top + \hat{U}_1\hat{\Lambda}_1\hat{U}_1^\top$$
$$\text{where, } \hat{u}_j \in \hat{U}_0 \text{ and } \lambda_{\min}(\hat{\Lambda}_0) - \lambda_{\min}(\hat{\Lambda}_0) \geq \frac{1}{4}\hat{\lambda}_j$$

Correspondingly, we can decompose $\mathbb{E}_{w_0} B_W^{\text{target}} w_0 w_0^\top (B_W^{\text{target}})^\top$ as the same shape

$$\mathbb{E}_{w_0} \left[ B_W^{\text{target}} w_0 w_0^\top (B_W^{\text{target}})^\top \right] = U_0 \Lambda_0 U_0^\top + U_1 \Lambda_1 U_1^\top$$

By using Davis-Kahn theorem, we have

$$\|U_1^\top \hat{u}_j\| \leq \|U_1^\top \hat{U}_0\| \leq \frac{\|U_1^\top H \hat{U}_0\|}{\frac{1}{4}\hat{\lambda}_j} \lesssim k\|H\|\kappa(B_W^{\text{source}})$$

Since

$$\|H\| \leq \bar{\gamma} + \left\| \mathbb{E}_{w_0} \left[ B_W^{\text{target}} w_0 w_0^\top (B_W^{\text{target}})^\top \right] - \mathbb{E}_{w_0} \left[ \hat{B}_W^{\text{target}} w_0 w_0^\top (\hat{B}_W^{\text{target}})^\top \right] \right\|$$

$$\leq 2 \left\| \mathbb{E}_{w_0} \left[ B_W^{\text{target}} w_0 w_0^\top (B_W^{\text{target}})^\top \right] - \mathbb{E}_{w_0} \left[ \hat{B}_W^{\text{target}} w_0 w_0^\top (\hat{B}_W^{\text{target}})^\top \right] \right\|$$

$$\leq 2 \left( 2k\sqrt{\frac{d_X}{n_1}} \|B_W^{\text{source}}\|^2 + 2\sqrt{\frac{k}{\dot{n}_{\text{target}}}} \right) \|B_W^{\text{target}}\|$$

then we have

$$\|U_1^\top \hat{u}_j\| \lesssim 8k \left( 2k\sqrt{\frac{d_X}{n_1}} \|B_W^{\text{source}}\|^2 + 2\sqrt{\frac{k}{\dot{n}_{\text{target}}}} \right) \|B_W^{\text{target}}\|\kappa(B_W^{\text{source}}) \leq \frac{1}{2}$$

which suggests $\|U_0^\top \hat{u}_j\| = \|[U_0, U_1]^\top \hat{u}_j - [0, U_1]^\top \hat{u}_j\| \geq 1 - \|U_1^\top \hat{u}_j\| \geq \frac{1}{2}$. Therefore, there exists some $u_m$ as one of the columns of $U_0$ that such $u_m^\top \hat{u}_j \leq \mathcal{O}(\sqrt{\frac{1}{k^*}})$. And therefore, we have

$$\hat{\lambda}_j \hat{u}_j^\top \left( B_W^{\text{source}}(B_W^{\text{source}})^\top \right)^{-1} \hat{u}_j \leq k^* \lambda_m(\hat{u}_m^\top \hat{u}_j)\hat{u}_j^\top \left( B_W^{\text{source}}(B_W^{\text{source}})^\top \right)^{-1} \hat{u}_j(\hat{u}_j^\top \hat{u}_m)$$

$$\leq k^* \lambda_m \hat{u}_m^\top \left( B_W^{\text{source}}(B_W^{\text{source}})^\top \right)^{-1} \hat{u}_m$$

$$\leq k^* \max_i \|W_i^*\|^2$$

$\square$

**Lemma B.7.**

$$\|W'W'^\top\|_* \leq \mathcal{O}\left( \frac{1}{\sigma_{\min}^2(B_W^{source})} \|B_W^{target} \mathbb{E}_{\nu_{target}}[w_0 w_0^\top](B_W^{target})^\top\|_* \right)$$

*Proof.*

$$\|W'W'^\top\|_* \leq \frac{1}{\sigma_{\min}^2(B_W^{\text{source}})} \|B_W^{\text{source}} W'(W')^\top (B_W^{\text{source}})^\top\|_*$$

$$\leq \frac{1}{\sigma_{\min}^2(B_W^{\text{source}})} \|\hat{B}_W^{\text{source}} W'(W')^\top (\hat{B}_W^{\text{source}})^\top\|_*$$

$$\leq \frac{1}{\sigma_{\min}^2(B_W^{\text{source}})} \|\hat{B}_W^{\text{target}} \mathbb{E}_{\nu_{\text{target}}}[w_0 w_0^\top](\hat{B}_W^{\text{target}})^\top\|_*$$

$$\leq \frac{1}{\sigma_{\min}^2(B_W^{\text{source}})} \|B_W^{\text{target}} \mathbb{E}_{\nu_{\text{target}}}[w_0 w_0^\top](B_W^{\text{target}})^\top\|_*$$

$\square$

## B.4 Sample complexity analysis – Formal version of Theorem 4.1

**Theorem B.3** (Formal theorem). *By running Algo. 2, in order to let $ER(\hat{\phi}_X, \nu_{target}) \leq \varepsilon^2$ with probability $1 - \delta$, where $\delta \geq (d_X k)^{10}$, then the number of source samples $n_{source}$ is at most*

$$\widetilde{\mathcal{O}}\left( \sigma^2(k^*)^2 \min\{\kappa(B_W^{source}, k^*\} \max_i \|W_j^*(i)\|_2^2 k d_X \varepsilon^{-2} \right)$$

$$+ \widetilde{\mathcal{O}}\left( \varepsilon^{-\frac{4}{3}} k^{\frac{7}{3}} d_W^{\frac{2}{3}} d_X \left( k^{\frac{2}{3}} d_W^{\frac{1}{3}} \underline{\sigma}^{-\frac{4}{3}} + \bar{\kappa}^2 \underline{\sigma}^{-\frac{2}{3}} \right) \right)$$

$$+ \widetilde{\mathcal{O}}\left( \bar{\kappa}^2 \sqrt{k} \left( k^2 d_X \bar{\kappa}^2 + \frac{d_W^{\frac{3}{2}}}{\underline{\sigma}^2} \sqrt{k + \log(d_W/\delta)} \right) \right)$$

Here $k^* = rank(\mathbb{E}_{w_0 \sim \nu_{target}} B_W w_0 w_0^\top B_W^\top)$ *represents the effective dimension of target and*

$$W_i^* = \underset{w \in \mathcal{W}_{source}}{\arg\min} \|w\|_2 \quad s.t \quad B_W^{source} w = u_i \sqrt{\lambda_i} \text{ where } U, \Lambda \leftarrow Eig(\mathbb{E}_{w_0 \sim \nu_{target}} B_W w_0 w_0^\top B_W^\top).$$

*as long as,*

$$\dot{n}_{target} \geq \widetilde{\Omega} \left( \varepsilon^{-\frac{4}{3}} (k^*)^{\frac{2}{3}} \left( d_W^{\frac{1}{2}} \underline{\sigma}^{-\frac{4}{3}} + k^{-\frac{2}{3}} d_W^{\frac{1}{6}} \bar{\kappa}^2 \underline{\sigma}^{-\frac{1}{3}} \right) \sqrt{k + \log(d_W/\delta)} \right)$$

$$n_{target} \geq \widetilde{\Omega} \left( (k + \log(1/\delta)) \varepsilon^{-2} \right)$$

*Proof.* By setting the target excess risk $\varepsilon^2$ and the generalization guarantees in Theorem B.2, we have

$$\sigma^2 \left( k d_X \log((\kappa N_i)/d_W) + \log\frac{1}{\delta} \right) k^* \epsilon_j^2 = \varepsilon^2 \tag{3}$$

After some rearrangement, we can directly have the guarantees for $n_1^j, n_0, \dot{n}_{\text{target}}, n_{\text{target}}$. Sum over the epoch gives our desired result. Now we will focus on $n_2^j$.

$$\begin{aligned}
n_2^j &\leq \widetilde{\mathcal{O}}(k^* \max_i \|W_j'(i)\|_2^2 \epsilon_j^{-2}) \\
&\leq \widetilde{\mathcal{O}}(k^* (\kappa(B_W^{\text{source}} + k^*)) \max_i \|W_j^*(i)\|_2^2 \epsilon_j^{-2}) \\
&\leq \widetilde{\mathcal{O}} \left( \sigma^2 (k^*)^2 \min\{\kappa(B_W^{\text{source}}, k^*\} \max_i \|W_j^*(i)\|_2^2 (k d_X + \log(1/\delta)) \varepsilon^{-2} \right)
\end{aligned}$$

where the first inequality comes from the definition and the second inequality comes from the Lemma B.6.

Finally, by union bounding on the $1 - \delta$ from Theorem B.2 and the event $\mathcal{E}_{\text{offline 1}}$ over all the epochs, we get the target result. $\qquad \square$

## B.5 Algorithms in [23] is a special case of Algo. 2

Specifically, in this paper, we aim to minimize.

$$(\hat{B}_W^{\text{target}} w_0)^\top (\hat{B}_W^{\text{source}} Q (\hat{B}_W^{\text{source}})^\top)^{-1} (\hat{B}_W^{\text{target}} w_0), \text{ where } \sum_i q_i = 1 \text{ [eq. 1]}$$

which can be equivalently written as

$$\text{trace}(V^\top Q V)^{-1} (V^\top u u^\top V) = \sum_i q_i^{-1} v_i^2$$

where $V$ comes from the eigendecomposition of $\hat{B}_W^{\text{source}} = UDV^\top$ and $u$ can be anything satisfying $\hat{B}_W^{\text{source}} v = \hat{B}_W^{\text{target}} w_0$.

Therefore, if we assume $q_i = |u_i|^\alpha / \sum_i |u_i|^\alpha$ for $\alpha > 0$, then [8] is equivalent to choosing $\alpha = 2$ and [23] is equivalent to choosing $\alpha = 1$. It is easy to see that $\alpha = 1$ is the optimal solution.

## B.6 More interpretation on results

**Lemma B.8.** *When = the target task is uniformly spread* $\|\mathbb{E}_{w_0 \sim \nu_{target}} B_W w_0 w_0^\top B_W^\top\| = \frac{1}{k}$ *and the task representation is well-conditioned* $\sigma_{\min}^2(B_W^{source}) = \frac{d_W}{k}$, *we have*

$$\|W_i^*\|_2^2 = \frac{1}{d_W}$$

*Proof.* Do a svd decomposition on the $B_W^{\text{source}}$ gives $\sqrt{\frac{d_W}{k}} U_1 V_1^\top$. For any $i$, let $w$ satisfies

$$\sqrt{\frac{d_W}{k}} U_1 V_1^\top w = \sqrt{\frac{1}{k}} u_i$$

Rearranging the above equality gives $V_1^\top w = \sqrt{\frac{1}{d_W}} U_1^\top u_i$. Because $W_i = \arg\min \|w\|_2$ satisfy the above constraints, we have

$$\|W_i^*\|_2^2 = \|V_1^\top w\|_2^2 = \left\|\sqrt{\frac{1}{d_W}} U_1^\top u_i\right\|_2^2 = \frac{1}{d_W}$$

$\square$

**Lemma B.9.** *Let*

$$\nu_1 = \arg\min_\nu \|\nu\|_1 \ s.t \ B_W \nu = B_W w_0$$

$$\nu_2 = \arg\min_\nu \|\nu\|_2 \ s.t \ B_W \nu = B_W w_0$$

*Then* $\|\nu_1\|_1^2 \geq \|\nu_2\|_2^2$.

*Proof.*

$$\|\nu_1\|_1^2 \geq \|\nu_1\|_2^2 \geq \|\nu_2\|^2$$

$\square$

# C    Results and analysis for target-agnostic

## C.1    Algorithm for target-agnostic

---
**Algorithm 3** Target-agnostic algorithm for benign source space
---
1: **Inputs:** Target probability $\delta, \bar{\kappa}, \underline{\sigma}$. Some constant $\beta_1, \beta_2, \beta_3$. Others same as Algo. 1.
2: Set $q_0$ as $q_0(e_t) = \frac{1}{d_W}, \forall t \in d_W$, and $q_0(w) = 0$ otherwise
3: Set $n_0 = \beta_1 \beta_1 \bar{\kappa}^2 \left(k^3 d_X \bar{\kappa}^2 + d_W^{\frac{3}{2}} \underline{\sigma}^{-2} \sqrt{k + \log(1/\delta)}\right)$. Collect $n_0 q_0(w)$ data for each task denoted as $\{Z_w\}_{w|q(w)\neq 0}$
4: Update $\hat{B}_X \leftarrow \mathcal{O}_{\text{offline 1}}^X(\{Z_w\}_{w|q_0(w)\neq 0})$ and $\hat{B}_W^{\text{source}} \leftarrow \mathcal{O}_{\text{offline}}^W(\{Z_w\}_{w|q_0(w)\neq 0}, \hat{B}_X)$
5: Compute $q_1$ as $q_1(v_i) = \frac{1}{k}, \forall i \in k$, and $q_0(w) = 0$ otherwise. Here $v_i$ is the $i$-th vector of $V$, where $U, D, V \leftarrow \text{SVD}(\hat{B}_W^{\text{source}})$
6: For any given budget $n_1$, collect $n_1 q_1(w)$ data for each task denoted as $\{Z_w\}_{w|q_1(w)\neq 0}$.
7: Update $\hat{B}_X \leftarrow \mathcal{O}_{\text{offline 2}}^X(\{Z_w\}_{w|q_1(w)\neq 0}), \hat{B}_W^{\text{source}} \leftarrow \mathcal{O}_{\text{offline}}^W(\{Z_w\}_{w|q_1(w)\neq 0}, \hat{B}_X)$
8: **Return** $\tilde{B}_X$
---

## C.2    Results and analysis

**Theorem C.1.** *In order to get* $\text{ER}(\hat{B}_X, \nu_{target}) \leq \varepsilon^2$, *we have w.h.p* $1 - \delta$, *source samples complexity is at most*

$$\widetilde{\mathcal{O}}\left(\frac{k^2 d_X Tr(B_W \mathbb{E}[w_0 w_0^\top] B_W^\top)}{\sigma_k^2(B_W^{source})}\varepsilon^{-2}\right) + \widetilde{\mathcal{O}}\left(\bar{\kappa}^2 \left(k^2 d_X \bar{\kappa}^2 + \frac{d_W^{\frac{3}{2}}}{\underline{\sigma}^2}\sqrt{k + \log(d_W/\delta)}\right)\right)$$

*as long as,*

$$\dot{n}_{target} \geq \widetilde{\Omega}\left(\varepsilon^{-\frac{4}{3}}(k^*)^{\frac{2}{3}}\left(d_W^{\frac{1}{2}}\underline{\sigma}^{-\frac{4}{3}} + k^{-\frac{2}{3}}d_W^{\frac{1}{6}}\bar{\kappa}^2\underline{\sigma}^{-\frac{1}{3}}\right)\sqrt{k + \log(d_W/\delta)}\right)$$

$$n_{target} \geq \widetilde{\Omega}\left((k + \log(1/\delta))\varepsilon^{-2}\right)$$

*Proof.* Again from Section B.2.1, we have w.h.p at least $1 - \delta$

$$\text{ER}(\hat{B}_X, \nu_{\text{target}}) \lesssim \sigma^2 n_{\text{target}} (kd_X + \log(1/\delta)) \text{Tr}\left(\left((B_W^{\text{source}})\left(\sum_{w\in\mathcal{S}} n_w ww^\top\right)(B_W^{\text{source}})^\top\right)^{-1} B_W \left(\mathbb{E}_{\nu_{\text{target}}} ww^T\right) B_W^\top\right)$$

$$+ \frac{k + \log(1/\delta)}{n_{\text{target}}}$$

then by using similar steps in Section B.2.2, we have

$$\text{Tr}\left(\left((B_W^{\text{source}})\left(\sum_{w\in\mathcal{S}} n_w ww^\top\right)(B_W^{\text{source}})^\top\right)^{-1} B_W\left(\mathbb{E}_{\nu_{\text{target}}} ww^T\right)B_W^\top\right)$$

$$\leq \|\square\|\text{Tr}(B_W\mathbb{E}[w_0 w_0^\top]B_W^\top)$$

$$\leq \frac{k}{n_1\sigma_k^2(B_W^{\text{source}})}\text{Tr}(B_W\mathbb{E}[w_0 w_0^\top]B_W^\top)$$

and therefore,

$$\text{ER}(\hat{B}_X,\nu_{\text{target}})\leq\widetilde{\mathcal{O}}\left(\frac{k^2 d_X}{n_1\sigma_k^2(B_W^{\text{source}})}\text{Tr}(B_W\mathbb{E}[w_0 w_0^\top]B_W^\top)\right)+\widetilde{\mathcal{O}}\left(\frac{k+\log(1/\delta)}{n_{\text{target}}}\right)$$

Rearranging the inequality gives the final bound. $\qquad\square$

### C.3 Compare to previous passive learning and the target-aware one

Again we want to compare this result with the previous one.

**Comparison with passive learning.** We first consider the cases in their paper that the target task is uniformly spread $\|\mathbb{E}_{w_0\sim\nu_{\text{target}}} B_W w_0 w_0^\top B_W^\top\| = \frac{1}{k}$. (See detailed setting in Section 4)

- When the task representation is well-conditioned $\sigma_{\min}^2(B_W^{\text{source}}) = \frac{d_W}{k}$. We have a passive one as $\widetilde{\mathcal{O}}(kd_X\varepsilon^{-2})$ while the target-agnostic active one $\widetilde{\mathcal{O}}(kd_X\frac{k^2}{d_W}\varepsilon^{-2})$.

- Otherwise, we consider the extreme case that $\sigma_{\min}^2(B_W^{\text{source}}) = 1$. We have passive one $\widetilde{\mathcal{O}}(d_X d_W\varepsilon^{-2})$ while the target-agnostic active one $\widetilde{\mathcal{O}}(k^2 d_X\varepsilon^{-2})$. Note this is better than the $\widetilde{\mathcal{O}}(k^3 d_X\varepsilon^{-2})$ in the target-aware case.

These two results indicate that when the targets are uniformly spread, target-agnostic AL can perform even better than the target-aware. But we want to emphasize that whether it is uniformly spread or not is unknown to the learner. Even $\kappa\left(\mathbb{E}_{w_0\sim\nu_{\text{target}}}[w_0 w_0^\top]\right) = 1$ can leads to ill-conditioned $B_W\mathbb{E}_{w_0\sim\nu_{\text{target}}}[w_0 w_0^\top]B_W^\top$.

We then consider the single target $w_0$ case.

- With well-conditioned $B_W$, the passive one now has sample complexity $\mathcal{O}(k^2 d_X\varepsilon^{-2})$ while the active gives a strictly improvement $\mathcal{O}(\frac{k^3 d_X}{d_W}\varepsilon^{-2})$.

- With ill-conditioned $B_W$ where $\sigma_{\min}(B_W) = 1$ and $\max_i\|W_i^*\| = 1$, that is, only a particular direction in source space contributes to the target. The Passive one now has sample complexity $\mathcal{O}(kd_X d_W\varepsilon^{-2})$ while our target-agnostic active one has $k^2 d_X\varepsilon^{-2}$.

These two results indicate that the target-agnostic approach gives a worse bound when the targets are not well-spread, which meets our intuition since the target-agnostic tends to learn uniformly well over all the levels. But it can still perform better than the passive one under the discrete case, which again indicates the necessity of considering the continuous setting.

**Save task number.** Again when ignoring the short-term initial warm-up stage, we only require maintaining $\widetilde{\mathcal{O}}(k)$ number of source tasks.

## D   Limitations from the theoretical perspective

Here we list some open problems from the theoretic perspective. We first list some room for improvements under the current setting

- **Not adaptive to noise $\sigma$:** From Section B.1.1, we get $\sin(\hat{B}_X, B_X)$ scales with the noise $\sigma$, which suggests less sample number $n_0, n_1$ is requires to get a proper estimation of $B_X$. In our algorithm, however, we directly treat $\sigma = \Theta(1)$ and therefore may result in unnecessary exploration.

- Bound dependence on $\min\{\kappa^2(B_X^{\text{source}}), k\}$: This extra dependence comes from the instability (or non-uniqueness) of eigendecomposition. For example, when $\mathbb{E}_{\nu_{\text{target}}}[B_W w_0 w_0^\top B_W^\top] = \frac{1}{k} I_d$, there are infinite number of eigenvector sets. On the other hand, given a fixed $B_W^{\text{source}}$, current methods of obtaining $W'$ are highly sensitive to the eigenvector sets from the target. A direct method is of course constructing a confidence bound around the estimated $\hat{B}_W^{\text{target}}$ and finding the best $W'$ under such set. But this method is inefficient. Whether there exists some efficient method, like a regularized optimization, remains to be explored in the future.

- **Require prior knowledge of $\bar{\kappa}, \underline{\sigma}$:** Finally, can we estimate and use those parameters during the training remains to be open?

Besides that specific problem, it is always meaningful to extend this setting into more complicated geometries and non-linear/non-realizable models. Specifically,

- **More complicated geometry.** One open problem is to get guarantees when $\mathcal{W}_{\text{source}}, \mathcal{W}_{\text{target}}$ is no longer a unit ball. (e.g., eclipse). Another problem is, instead of considering the geometry of $\mathcal{W}$, we should consider the geometry of $\psi_W(\mathcal{W})$.

- **Nonlinear models.** Consider nonlinear $\phi_X, \phi_W$ is always challenging. In [9, 10], they provide some guarantees under the passive by using kernel methods or considering a general model. Can we extend this to the active setting?

- **Non-realizable model.** Like many representation learning papers, we assume the existence of a shared representation, which suggests more source tasks always help. In practice, however, such representation may not exist or is more over-complicated than the candidate models we assume. Under such a misspecification setting, choosing more tasks may lead to negative transfer as shown in Figure 5 in the experiments. Can we get any theoretical guarantees under such a non-realizable setting?

# E  Experiment details

Here we provide detailed settings of three experiments – synthetic data, pendulum simulator, and the real-world drone dataset, as well as more experimental results as supplementary. All the experiments follow a general framework proposed in Section 3 with different implementation approaches according to different settings, which we will specify in each section below. Note that in all these experiments, we only focus on a single target.

## E.1  Synthetic data

### E.1.1  Settings

|  | bilinear | nonlinear $\psi_X$ | nonlinear $\phi_X$ |
|---|---|---|---|
| target number | 800, 8000 | 800, 8000 | 800, 8000 |
| $d_X$ | 200 | 10 | 20 |
| $d_{\psi_X}$ | 200 | 200 | 20 |
| $d_W$ | 80 | 80 | 80 |
| $k$ | 4 | 4 | 4 |
| $\phi$ structure | random matrix | random matrix | MLP with layers [20, 20, 4] |
| inputs distribution | $\mathcal{N}(0, I)$ | $\mathcal{N}(0, I)$ | $\mathcal{N}(0, I)$ |
| label noise variance | 1 | 1 | 1 |

Table 3: Model used to generate the synthetic data.

**Data generation**  We show the model and corresponding parameters used to generate the synthetic data in Table. 3. Some additional details include, 1) When generating random matrix $B_X$ for bi-linear and unknown non-linear $\psi_X$, we tried different seeds (denoted as *embed_matrix_seed* in the codes) and deliberately make the matrix ill-conditioned (so $\kappa(B_W)$ is large). Because most of them behave similarly so we only present partial results here. 2) When generating random MLP for nonlinear $\phi_X$, we only use the unbiased linear layer and ReLU layers.

In the main paper Table 5.2, we use target number = 8000 cases to show more contrast.

The nonlinear Fourier feature kernel $\psi_X$ is defined as $\psi_X(x) = \cos(Ax + B)$, where $A \in \mathbb{R}^{d_{\psi_X} \times d_X}, B \in \mathbb{R}^{d_{\psi_X}}$ and each entry of $A, B$ is i.i.d. Gaussian.

**Training models and optimizer**   Here we state the details of the model used during the learning, which might be different from the model used to generate the data. Specifically, for the bi-linear and unknown non-linear $\psi_X$, we use the exact $\mathbb{R}^{d_{\psi_X} \times k}$ matrix structure as stated in the theorem. For the nonlinear $\phi_X$, we use a slightly larger MLP with layers [20, 20, 20, 4] compared to the model used to generate the data to further test the adaptivity of our algorithm since the exact underlying structure of MLP is usually unknown in reality. As for the joint training approach, we use Adam with $lr = 0.1$ for the bi-linear and unknown non-linear $\psi_X$, and SGD with $lr = 0.1$ for nonlinear $\phi_X$ as the optimizer (The learning rate is large because this is an easy-to-learn synthetic data) We mixed all the target and source data and do joint GD-based methods on them. Notice that the goal for those experiments is not to achieve the SOTA but to have a fair comparison. So all those hyper-parameters are reasonable but not carefully fine-tuned.

**Detailed implementation for AL strategy**   Both the input space $\mathcal{X}$ and the task space $\mathcal{W}$ of synthetic data lie perfectly in a ball and the underlying model is linear in terms of $w$. Therefore, we can use the almost similar algorithms as proposed in Algo 2 for target-aware and Algo. 3. We slightly adjust parameter dependence on $d_X, d_W, k$ but the general scaling between different stages in each epoch remains the same. Another difference is that, instead of using the MLLAM as specified in Section B.1.1, we do a joint-GD since the implementation of MLLAM in a non-idealistic setting (nonlinear $\phi_X, \psi_X$ is unclear and challenging.)

**Metrics**   We consider the worst-case distance between ground truth and estimate columns space $U, \hat{U}$ as $\text{dis}(U, \hat{U}) = \min_u \|u_i^\top \hat{U}\|_2$. Such distance will be used in both computing the similarity between ground truth and estimated input space $B_W, \hat{B}_W$. In addition, it will also be used in measuring the change of $q_2$ across each epoch so we can save task numbers by maintaining the same $q_2$ as long as the change is small, which we will specify in the next paragraph.

**Saving task number approach.**   In addition to the comparison between target-agnostic AL, target-aware AL, and the passive, we also consider the saveTask case, where we reduce the number of times recomputing the $q_1$. Specifically, we denote $W_{j-1}, W_j \in \mathbb{R}^{d_W \, times \, k}$ as the exploration source tasks in the previous and current epoch. And only switch to the new target-agnostic exploration set when $\text{dis}(\text{rowSpace}(B_{j-1}), \text{rowSpace}(B_j)) \leq 0.8$ where $0.8$ is some heuristic threshold parameter.

### E.1.2 Results

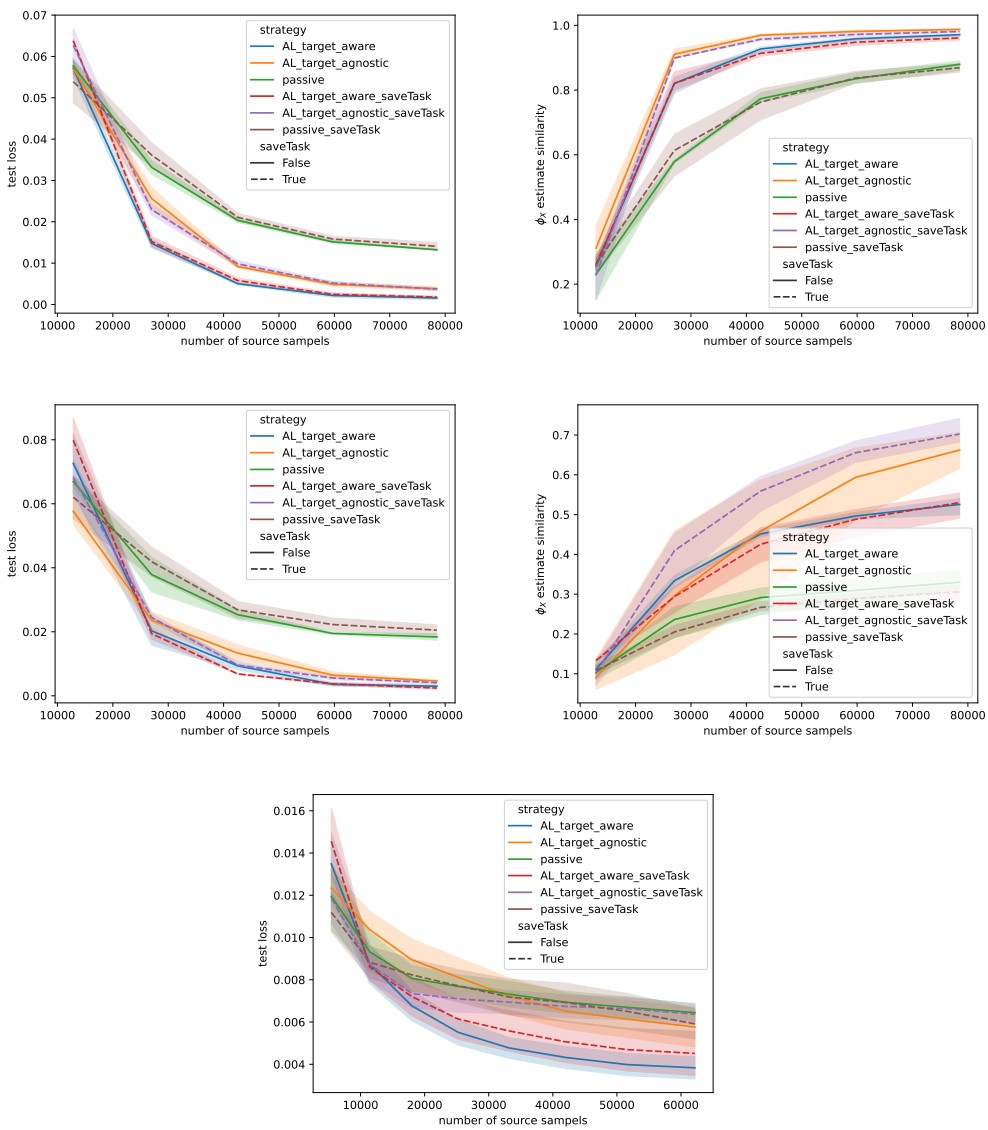

Figure 3: **Results on synthetic data with 8000 target sample** Left side presents the test loss and the right side presents the similarity between the column space of the ground truth $\phi_X$ and the estimated $\hat{\phi}_X$. Notice that how to measure the similarity on neural networks is unclear so we skip this result. **Top and middle:** Results of the nonlinear kernel. The target-aware AL gets the lowest test loss while the passive gets the highest. In terms of saveTask, we notice that reducing task switch number does not affect the performance a lot. From the left figure, the target-agnostic AL gets the best estimation which aligns with our design intuition that target-agnostic AL should have a universal good estimation in all directions. It is a little surprising to us that the passive one performs worst. We conjecture the reason that the GD-based oracle is not that good for joint-task training and should again have better performance when using [12, 10]. **Bottom:** Result of non-linear representation. Here we notice that the saving task strategy leads to slightly worse performance. While the target-aware AL still gives the worst test loss, the difference between passive and target-agnostic AL is small due to the complexity of the shallow net.

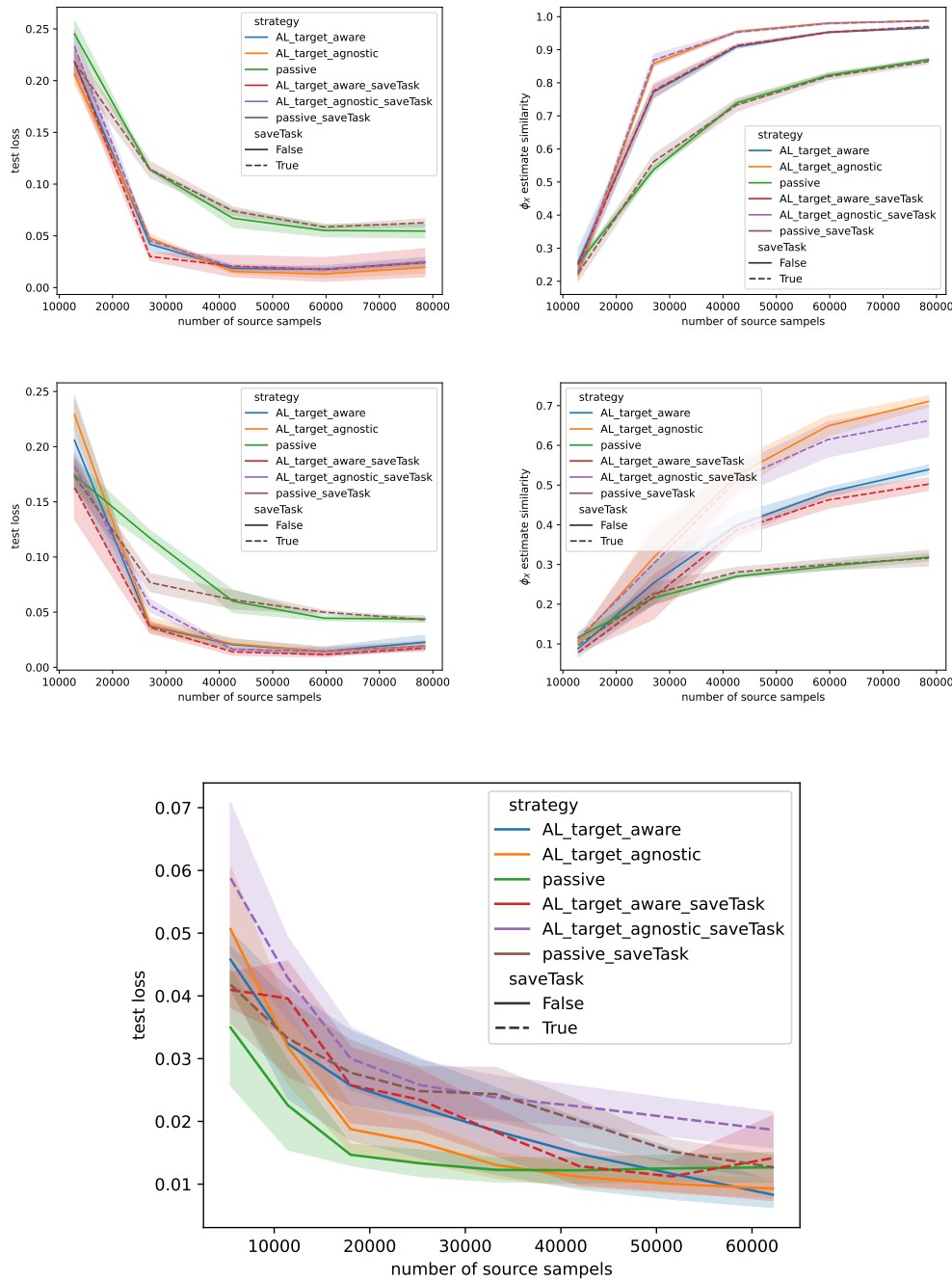

Figure 4: **Results on synthetic data with 800 target sample Top and middle:** The bilinear and nonlinear $\psi_X$ case gives a similar performance as before. **Bottom:** For $\phi_W$ as a neural net, we notice here the AL does not show an advantage until the very end where the passive stops decreasing. This may suggest for nonlinear representation, more target data may be needed for a beneficial source selection compared to the bilinear $\phi$.

### E.2 Pendulum simulator

#### E.2.1 Settings

**Data generation**    We consider the following continuous-time pendulum dynamics model adopted from [24]:

$$ml^2\ddot{\theta} - ml\hat{g}\sin\theta = u + f(\theta, \dot{\theta}, w)$$

where $\theta, \dot{\theta}, \ddot{\theta}, u$ are angle, angular velocity, angular acceleration, and control, $m, l, \hat{g}$ are mass, pole length, and the gravity estimation, and finally, $f$ is the unknown residual dynamics term to be learned with $w$ the environment parameter. The ground truth $f$ is given by

$$F = \|R\|_2^2 \cdot R, R = c - \left[ \begin{array}{c} l\dot{\theta}\cos\theta \\ -l\dot{\theta}\sin\theta \end{array} \right]$$

$$f(\theta, \dot{\theta}, w) = \underbrace{\vec{l} \times F}_{\text{air drag}} - \underbrace{\alpha_1\dot{\theta} - \alpha_2\dot{\theta}|\dot{\theta}|}_{\text{damping}} + \underbrace{ml(g - \hat{g})\sin\theta}_{\text{gravity mismatch}}$$

$$w = [c_x, c_y, \alpha_1, \alpha_2, \hat{g}, 0 \text{ or } 1]$$

where $c = [c_x, c_y]$ is external wind, $\alpha_1, \alpha_2$ are damping coefficients and $g$ is the true gravity.

We let $x = [\theta, \dot{\theta}]$ denote the input to $f$. Notice here the last element of $w$ is a dummy feature. For the source tasks, we always have $w[6] = 0$ since all the source parameters are known. For the single target task, we have $w_{\text{actual\_target}}$ to generate the data, so $w_{\text{actual\_target}}[6] = 0$. But the learner only observes the $w_{\text{target}} = [0, 0, 0, 0, 0, 1]$, which indicates the unknown environment of the target. In the simulator, we collect data using a stochastic policy to approximate i.i.d. data distribution.

It is easy to see that $f$ is highly nonlinear regarding $x, w$. Therefore we use the known nonlinear feature operator $\psi$ to make it close to the linear model with some misspecification:

$\psi_X$ is the Fourier feature kernel which has been defined in the synthetic data section

$$\psi_W(w) = [l_x, l_y, g, \alpha_1, \alpha_2, CxCy, Cx^2, Cx^2Cy, C_x^3, Cy^2, Cy^2C_x, C_y^3, 0 \text{ or } 1]$$

Other common parameters are specified in Table. 4.

| target number | $d_X$ | $d_{\psi_X}$ | $d_W$ | $d_{\psi_W}$ | $k$ | $\phi$ structure | inputs distribution | label noise variance |
|---|---|---|---|---|---|---|---|---|
| 4000 | 2 | 60 | 13 | 6 | 8 | bilinear | (See details above) | 0.5 |

Table 4: Model parameters for pendulum simulator.

**Training models and optimizer**    We again use the bilinear model. For the training methods, we first do joint-GD as before using AdamW with $lr = 0.01, wd = 0.05, \text{batch\_size} = 512$. Then after joint training, we freeze the $\phi_X$ parts and only trained on the targets to get the non-shared embed $\phi_W(w_{\text{target}})$. Another modification is that, since we are in the misspecification setting, using data collected in stage 3 might amplify the errors when estimating the target-related source. To tackle this negative transfer learning, we only use the data collect from stage 2 in previous the epochs to compute $q_3$. While in the synthetic data, all data, including one from stage 3, collected in previous epochs can be used.

**Detailed implementation for AL strategy**    The input space $\mathcal{X}$ and task space $\mathcal{W}$ of this pendulum data again lie perfectly in a ball after some normalization. Nevertheless, the underlying model is no longer linear in terms of $w$, which adds some extra difficulties to the optimal design on $w$. Here we use the adaptive sampling methods mentioned in the main paper. That is, we will iteratively sample from $\mathcal{W}_{\text{source}}$ and find the ones that minimize follows.

$$\min_{\{w_i\} \in \mathcal{W}_{\text{source}}} \|\hat{B}_{W,j}^{\text{source}}\psi_W(w_i) - u_i\sqrt{\lambda_i}\|$$

where $u_i\sqrt{\lambda_i}$ is defined in line 9. Other parts of the algorithm can still be implemented as in the synthetic data section.

**Using learned $f$ for control**  To show that a better dynamics model can transfer to better control performance, we deploy the following nonlinear controller $\pi(x, \hat{f})$ as a function of $\hat{f}$ (prediction result of $f$ in the target task):

$$u = -ml\hat{g}\sin\theta - \hat{f}(\theta, \dot{\theta}) - ml^2(K_P\theta + K_D\dot{\theta})$$

Here we focus on the regulation task, i.e., $\|x\| \to 0$. It is worth noting that the above controller is guaranteed to be exponentially stable: $\|x\| \to \eta$ exponentially fast, where $\eta$ is an error ball whose size is proportional to $\|f - \hat{f}\|_\infty$.

### E.2.2   Results

In the main paper, we use the unobservable actual target as $[0, 0, 1, 0.5, 0, 0]$. Here we give more results in Figure. 5

### E.3   Real-world drone flight dataset

### E.3.1   Settings

**The training model and optimizer**  Here we use two layer MLP model as specified below. For the training methods, we do joint-GD as before using AdamW with $lr = 0.005$ and batch_size$= 1000$. Other common parameters are specified in Table. 5.

| target number | $d_X$ | $d_{\psi_X}$ | $d_W$ | $d_{\psi_W}$ | $k$ | $\phi$ structure |
|---|---|---|---|---|---|---|
| 500 | 11 | 11 | 18 one-hot | 18 | 2 | MLP with hidden layers $[11, 2]$ |

Table 5: Model parameters for drone dataset.

**Data generation**  We use the same data as stated in the main paper.

**Detailed implementation for AL strategy**  Unlike the previous two settings where the task space $\mathcal{W}$ is continuous, here we consider a discrete task space. Therefore the Algo. 2 no longer works. Therefore, here we use a similar technique as the Algorithm proposed in [8], which can be seen as a special case under the general Algo. 1. We want to emphasize that this choice is due to the limitation of real-world datasets, i.e., we can not arbitrarily query $w$ to sample, and the main purpose is to show the potential of such a framework in real-world robotics applications.

### E.3.2   Results

In the main paper, we provide the result when assuming a bilinear underlying model. Here we further show the effectiveness of our methods under nonlinear $\phi_X$.

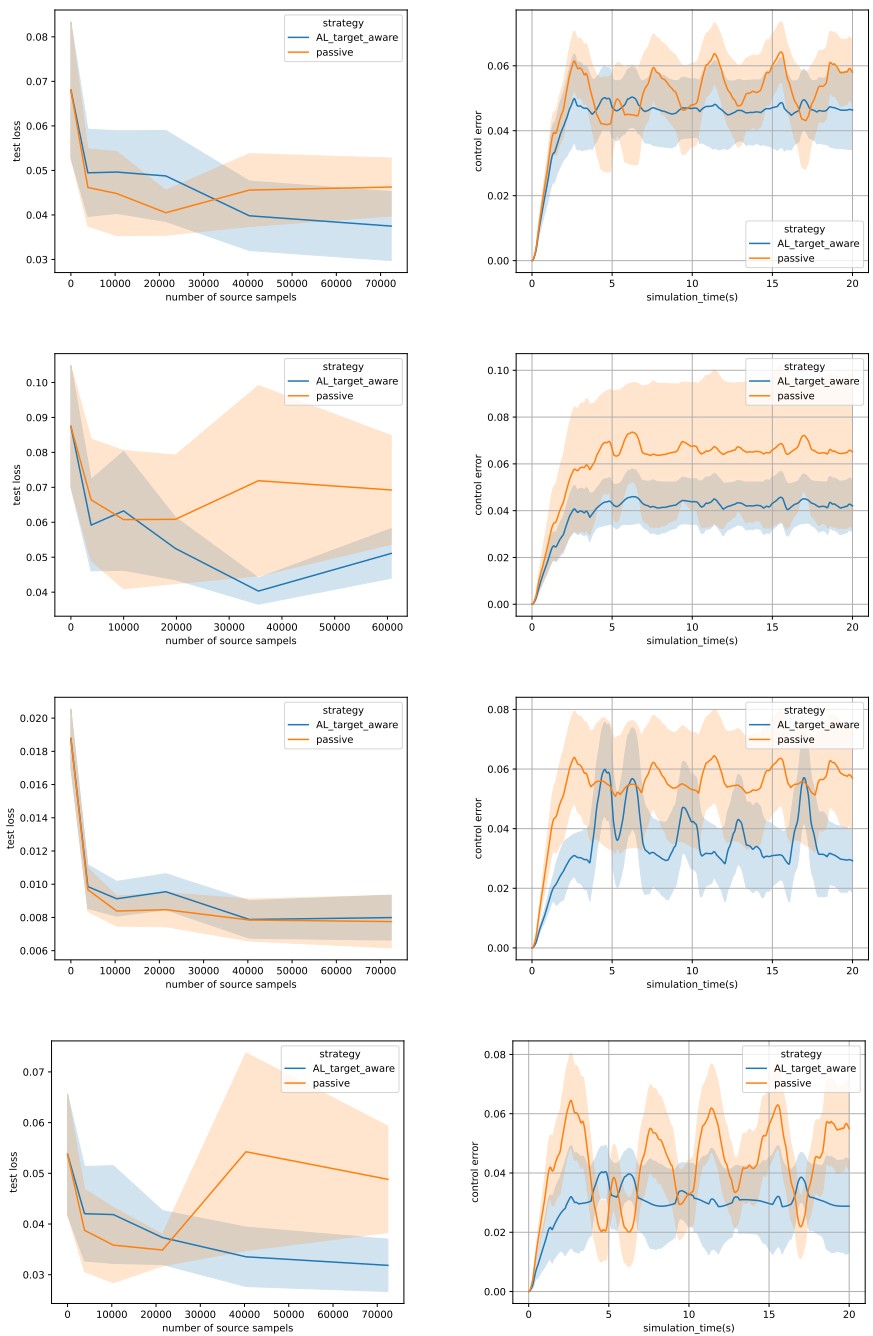

Figure 5: **Results on pendulum simulator for a specific target**. **Left:** The test loss of the estimated model $\hat{f}$. The passive strategy suffers from negative transfer while the active strategy steadily decreases. **Right:** The control error using final output $\hat{f}$. Here we use a model-based nonlinear policy $\pi(x, \hat{f})$. The model learned from active strategy leads to better control performance. From top to bottom, we have the unobservable $w_{\text{actual\_target}}$ as $[0, 0, 0.5, 0, 0.5, 0], [0, 0, 1, 1, -1, 0], [0, -1, 0.5, 0, 0.5, 0], [0, 0.1, 0, -1, 0.5, 0]$. Overall, although AL does not always have a dominating advantage, most times it is more stable and can gain better test loss at the end.

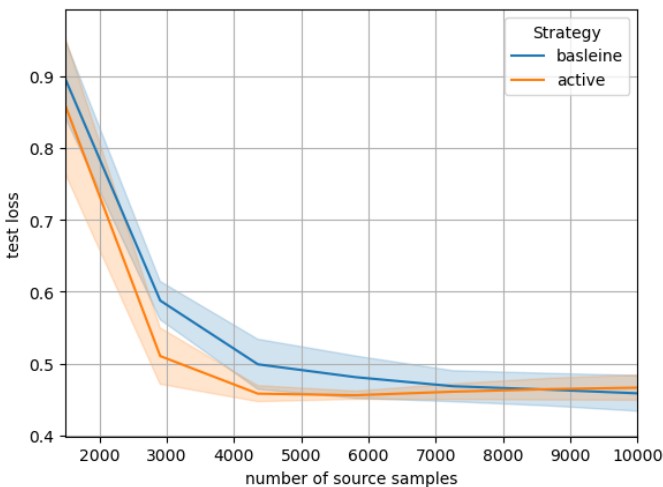

Figure 6: **Results on the real drone dataset** with target `drone_type_A_30_z` by using a neural net model. Our active strategy could converge faster than the passive strategy in the neural net model setting. Active strategy is able to converge faster than uniform sampling with smaller variances in the latter stage.

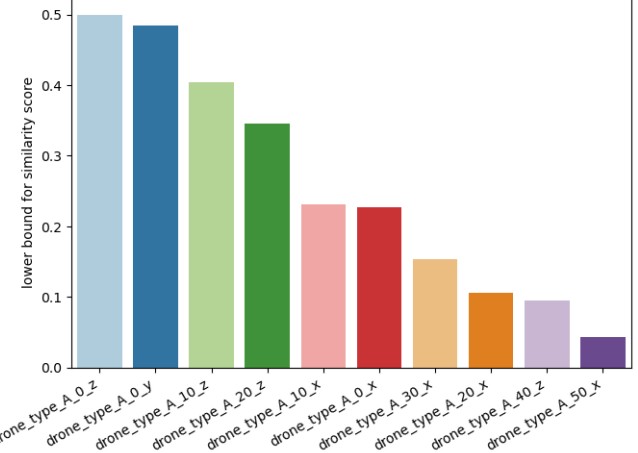

Figure 7: Top 10 the most similar source tasks. Again, given the target environment, the algorithm successfully finds the other `drone_type_A` environments as relevant sources, which aligns with our observation in the main paper.