# OpenReview forum: "Active representation learning for general task space with applications in robotics"
_NeurIPS.cc/2023/Conference — NeurIPS 2023 poster_

### Official Review · Reviewer_7KMj · 2023-06-30

**Soundness:** 3 good
**Presentation:** 1 poor
**Contribution:** 2 fair
**Rating:** 5
**Confidence:** 2

**Summary:**

The paper introduces an algorithm for active representation learning in multi-task settings. The algorithm generalizes to settings in which the “task space” is either discrete or continuous.

The overall setting involves an (optionally) nonlinear map between an input space X and an intermediate feature space via $\psi_X$ and $\phi_X$, that then undergoes a final multiplication with “task features” obtained by applying $\psi_W$ and $B_W$ to task parameters $w$. The goal is to learn $\hat{\phi}_X$ and $B_W$ given the possibility of actively selecting tasks $w$ within a “source” domain, and evaluating on tasks in a “target” domain.

The proposed approach can work for both “coarse exploration”, “fine target-agnostic exploration” and “fine target-aware exploration”.

The authors provide a theoretical analysis for the “benign setting” in which the source task space satisfies specific assumptions. Moreover, they provide experimental results beyond this setting, and on some real-world datasets.


**Strengths:**

The paper is heavily theoretical in nature, and seeks to formally introduce the problem with precise assumptions, also offering theoretical guarantees given a “benign” version of the problem setting.

The experimental results show that the target-aware method indeed beats the target-agnostic method by a large margin, giving credence to the significance of the target-aware method.


**Weaknesses:**

Overall, the paper is very hard to parse, and seems to take much background as granted. In the preliminaries, a very large number of assumptions are made on the problem setting. It is unclear given my limited expertise in the subfield whether all these assumptions undermine the significance of the approach or not. It would be useful to more thoroughly compare the assumptions and design choices with the literature on the topic, to better contextualize what the paper brings to the table and what the contribution is.

The setting itself seems interesting, but not enough emphasis is put on why it is significant and worthwhile exploring. It seems that the approach effectively requires us to be able to at-will sample tasks within their “task parameter space”, which seems to be extremely hard to do for anything beyond simulation. The drone experiment seems to do away with this by defining the parameter space as a one-hot vector, but not enough emphasis is placed on the significance of such a choice.

Furthermore, the results on the drone dataset experiment seem to show that the proposed method converges faster, but ultimately reaches the same accuracy as the passive strategy.


**Questions:**

I would like the authors to address the points raised in “Weaknesses”, in order to better clarify my uncertainty related to the paper’s significance.

Edit: increased the score after taking into account the authors' reply.

**Limitations:**

The authors address in the appendix some limitations related to the theoretical analysis. It would however be helpful if they could list some limitations from the point of view of how related the setting is to real-world multi-task learning problems.

---

> ### Author Rebuttal · Authors · 2023-08-09
>
> ### Q1: The setting itself seems interesting, but not enough emphasis is put on why it is significant and worthwhile exploring. It seems that the approach effectively requires us to be able to at-will sample tasks within their “task parameter space”, which seems to be extremely hard to do for anything beyond simulation. The drone experiment seems to do away with this by defining the parameter space as a one-hot vector, but not enough emphasis is placed on the significance of such a choice.
>
> **About the feasibility of “at-will sample”**:  According to my understanding, the infeasibility may comes from two perspectives: 1\ The computationally tractable task parameter space is too idealistic  (e.g. ball, pure linear) that might not even exist in nature. 2\ The required sample parameter exists but is hard to access due to many practical reasons.
>
> Regarding 1). In most cases, using a ball is a good approximation to the parameter space. Moreover, there always exists other approximation methods. For example, in this paper, we show adaptive sampling in this paper experiment 2: pendulum simulation. (initially mentioned in line 194-196 “harder geometry). Specifically, we first assume the augmented task parameter $\psi_W(w)$ lies everywhere in the ball, therefore we can quickly solve a targeted “fake” $\psi_W(w)$ whose corresponding $w$ might not exist. Next we use the method described in line 735-741 to iteratively find the closest projection onto the feasible space. The effectiveness of our method has been verified by the result.
>
> Regarding 2), we admit that there is no definitive answer at this point since the continuous hardware needs real-world interaction. We want to emphasize that this work focuses on initial theoretical validity. In addition, we conduct the pendulum simulation, which paves the way for the real-world interactive setting in the next step.
>
> **About the drone experiment**: While our experiment is ultimately designed for real-world interactive data collection, it is a common approach to start with an existing dataset as a preliminary verification step. (e.g. In usual pool-based active learning, they are aimed to simulate the interactive data annotation system but their experiments are mostly conducted on existing datasets.)
>
> Therefore, one of the reasons for using one-hot vectors is limited by the existing datasets, which are collected in a discrete way. But we admitted that this cannot help show the effectiveness of our algorithm under continuous space setting, therefore we add experiment 1: synthetic and experiment 2: pendulum simulation.
>
> ### Q2: Furthermore, the results on the drone dataset experiment seem to show that the proposed method converges faster, but ultimately reaches the same accuracy as the passive strategy.
>
> It is as expected.In our setting, the asymptotic performance of passive and active learning is supposed to be the same. Specifically, since we first learn a joint representation from the source tasks and then do a transfer learning on a few-shot target task, the final excess risk not only *depends* on the source sample complexity but also on *target sample complexity*. (see Line 242 in this paper, Eqn 18 in [1] where they consider the discrete case) As we shown in Table 5, the target number is 500, so the active and passive will finally converge to the same error level as long as they use the same number of target data.
>
> Notice that according to line 256-262 in our paper (continuous case) and Theorem 3.10 in [1] (discrete case), the $\text{source sample number} / \text{target sample number radio} \approx d_{\psi_W}$ when the target error stops decreasing. Because the other two tasks have larger $ d_{\psi_W}$, 200 and 60, so their results didn’t reach their limits. But when it comes to the real-word data, our choice is relatively limited.
>
> ### Q3: Overall, the paper is very hard to parse, and seems to take much background as granted. In the preliminaries, a very large number of assumptions are made on the problem setting. It is unclear given my limited expertise in the subfield whether all these assumptions undermine the significance of the approach or not. It would be useful to more thoroughly compare the assumptions and design choices with the literature on the topic, to better contextualize what the paper brings to the table and what the contribution is.
>
> A: We are sorry for confusion, please refer to our responses to all. We will add those discussions in the extra page in revision.

---

> > ### Comment · Reviewer_7KMj · 2023-08-11
> > **Reply to Authors**
> >
> > I thank the authors for their reply. Overall, I still have concerns with respect to real world applicability of the method, specifically related to point 2) that has been raised in this reply (how to even sample from the task space at will in real world applications).
> > However, due to their efforts in clarifying mine and other reviewers' points, I am willing to increase my score, albeit remaining at low confidence.

---

> > > ### Author Response · Authors · 2023-08-17
> > >
> > > Thanks for you comment.
> > >
> > > Maybe add on to the "sampling from the task space" part. One example is in the drone datasets. In reality, each task (environment) parameter can be considered as the some exact windspeed, which can be simulated by Caltech Real Weather Wind Tunnel as shown in https://www.science.org/doi/10.1126/scirobotics.abm6597.  But in this paper, as a *preliminary* step, we only validate our algo on the already collected offline datasets instead of implementing real online interactive collection, which is costly. That's why we formulate that  drone experiment as one-hot vector in our paper.
> > >
> > > Specifically, if we do real-world experiment, then we aims to learn $\phi_{\psi_W}$. While on collected dataset, we aimed to learn  $\phi_{\psi_W}(w_i)$ for $i \in [\text{total num of discrete tasks}]$. So if $\{w_i\}$ is really spread out, then it is equivalent to learning $\phi_{\psi_W}$. Overall, these two are not exactly the same but can be still be a relevant experiment support.
> > >
> > > Nevertheless, we said there is no definitive answer because the feasibility of implementation largely depends on the specific robotic task you try to apply on. We acknowledge that, as a general framework, it might be challenging to implement for tasks where establishing a real physical environment is either costly or inaccessible.

---

### Official Review · Reviewer_7vYB · 2023-07-06

**Soundness:** 4 excellent
**Presentation:** 3 good
**Contribution:** 3 good
**Rating:** 7
**Confidence:** 4

**Summary:**

This work considers the multi-task learning problem and proposes an algorithm to select related source tasks to tackle a known target task in an active learning like setup. The analysis is primarily for the linear function classes and handles discrete and continuous task spaces. The proposed algorithm operates in 3 stages: (1) a coarse sampling of the tasks (2) More efficient target-agnostic sampling of diverse tasks (3) target-aware sampling from a sub-space of related tasks. The 2nd and 3rd stage in particular allow for tighter sample complexity bounds since it allows us to efficiently sample from the task space. As a result, the sample complexity bounds depend on $k*$ which is the relationship between the target space and the relevant source space. Finally, this algorithm is used in three scenarios and an active learning strategy is shown to outperform a passive one.


**Strengths:**

The paper presents a theoretically motivated algorithm to sample source tasks that are relevant to the target task. While the theme of active sampling is not entirely new, few of them provide algorithms with sample complexity bounds. While there are many gaps to fill (linear function classes, isotropic Gaussian inputs), this paper takes a step in the right direction.

There are two interesting ideas in this work. The first is to reduce the number of tasks that we consider by selecting a diverse set of tasks (subspace $V$) that spans the task space. This is the primary role of stage 2 or the task-agnostic exploration. The second idea (similar to Yifang et al.) is to perform a task-aware search of the task-space and only use source tasks that are relevant to the target task. This improves sample complexity and the benefit is stronger if a smaller subspace of source tasks is strongly related to the target task.

The paper presents many results on 3 different experimental scenarios that primarily compare passive multi-task and active target-aware learning with clear sample complexity benefits for the latter.


**Weaknesses:**

**Experimental setup**: To the best of my understanding, all the experiments in the manuscript (including appendix) compare passive to target-aware active learning strategies. How does the proposed method compare to a target-aware strategy that doesn't use stage 2 and only uses stage 3. I understand the theoretical benefit of stage 2 is clear and it allows us to work with fewer source tasks but is this reflected experimentally?

**Improving the discussion on related work:** It would be nice to see the authors discuss some related work on multi-task learning and task-relatedness. This include some older work on passive multi-task learning [1,2], learning theory [3,4,5], task-distances [6,7,8], task-grouping [10,11] and weighted-training [6,12]. There are a lot more works that I haven't listed out that I think are very relevant to this work.


1. Caruana, Rich. "Learning many related tasks at the same time with backpropagation." Advances in neural information processing systems 7 (1994).
2. Baxter, Jonathan. "Learning internal representations." Proceedings of the eighth annual conference on Computational learning theory. 1995.
3. Ben-David, Shai, et al. "A theory of learning from different domains." Machine learning 79 (2010): 151-175.
4. Hanneke, Steve, and Samory Kpotufe. "On the value of target data in transfer learning." Advances in Neural Information Processing Systems 32 (2019).
5. Crammer, Koby, Michael Kearns, and Jennifer Wortman. "Learning from Multiple Sources." Journal of Machine Learning Research 9.8 (2008).
6. Thrun, Sebastian, and Joseph O'Sullivan. "Discovering structure in multiple learning tasks: The TC algorithm." ICML. Vol. 96. 1996.
7. Zamir, Amir R., et al. "Taskonomy: Disentangling task transfer learning." Proceedings of the IEEE conference on computer vision and pattern recognition. 2018.
8. Achille, Alessandro, et al. "Task2vec: Task embedding for meta-learning." Proceedings of the IEEE/CVF international conference on computer vision. 2019.
9. Standley, Trevor, et al. "Which tasks should be learned together in multi-task learning?." International Conference on Machine Learning. PMLR, 2020.
10. Ramesh, Rahul, and Pratik Chaudhari. "Model Zoo: A Growing" Brain" That Learns Continually." arXiv preprint arXiv:2106.03027 (2021).
11. Fifty, Chris, et al. "Efficiently identifying task groupings for multi-task learning." Advances in Neural Information Processing Systems 34 (2021): 27503-27516.
12. Chen, Shuxiao, Koby Crammer, Hangfeng He, Dan Roth, and Weijie J. Su. "Weighted training for cross-task learning." arXiv preprint arXiv:2105.14095 (2021).




**Questions:**

It seems like an important part of the algorithm is to use $g(f, A)$ from Equation 2, which estimates which estimates the most relevant source tasks. How would this change if were to use non-linear function classes. In general, I was wondering which parts of the algorithm rely on the function being linear.

In practice, the sample complexity bounds may be too pessimistic (which isn't easy to get around). How do we decide the number of samples to select in each round of active sampling and what are some other key hyper-paramters? It feels like the additional complexity of the methods leads to a lot more hyper-parameters which seems like it will be harder to get it to work in practice.

The paper has dense notation dense but I appreciate the authors effort to unpack all the results. While I was able to follow the paper, the notation can be hard to follow at times. It may help the reader if some parts are simplified in the main paper. For example the algorithm could be written without all the precise details needed to achieve the sample complexity bound.


**Limitations:**

The paper discusses some of the limitations in Appendix C in sufficient detail.

---

> ### Author Rebuttal · Authors · 2023-08-09
>
> ### Q1.1: Experimental setup: To the best of my understanding, all the experiments in the manuscript (including appendix) compare passive to target-aware active learning strategies.
>
> We want to correct the in our experiment 1: synthetic data, we provide comparison between passive and target-agnostic active learning strategies in Figure 3 and 4 in Appendix D.1.2. As expected, the target-agnostic AL performs similar or worse than target-aware one in terms of the excess risk on the target task, but obtains better overall model estimation, which implies it should perform better on other unaware targets.
>
> We don’t provide the target-agnostic experiment 2:pendulum, experiment3: drone because the target-agnostic is designed for the case where $d_{\psi_W} >> k$.
>
> ### Q1.2 How does the proposed method compare to a target-aware strategy that doesn't use stage 2 and only uses stage 3. I understand the theoretical benefit of stage 2 is clear and it allows us to work with fewer source tasks but is this reflected experimentally?
>
> To clarify what you mean for “stage 1 + stage 3”, we understand that as follows: instead of doing $\epsilon^{-\frac{4}{3}}$ target-agnostic exploration on the $O(k)$-subspace, we do exploration on the whole $d_{\psi_W}$ space, and then do target-aware exploration.
>
> In that case, in the experiment 1: synthetic data, the “error vs number of source samples” will be similar to the case using all three stages. But we need to work on $O(d)$ number of source tasks after the warmup stage, which largely increases the source tasks number we need to work to, as expected.
>
> On the other hand, experiment 2: pendulum simulation and experiment 3: drone can not save much from stage 2 because of  $d_{\psi_W} ~=O(k)$, as expected. Therefore, as suggested in Line 7 in Algorithm 1, in these two experiments, we did skip stage 2.
>
> But if you are suggesting we only do $\epsilon$-independent fix length warm-up stage once. Then we think it is unlikely to get any good results unless it’s in very special cases - the target-relevant sources are very sparse.
>
> ### Q2: Improving the discussion on related work
>
> Thanks for pointing this out. While some are different to our setting, we should definitely add a related work section for discussion. + standard (passive) multitask learning. Please refer to “literature review” in our comment section for more details.
>
> ### Q3: It seems like an important part of the algorithm is to use from Equation 2, which estimates the most relevant source tasks... n general, I was wondering which parts of the algorithm rely on the function being linear.
>
> If *not* considering the computational efficiency, then it is possible for $f$ in $g(f,A)$ be non-linear $\phi_W$ as long as the structure between $\phi_W$ and $\phi_X$ is linear. The motivation of this optimal design idea is detailed in Appendix A.2.1. Specifically, by rearranging the previous passive learning bound, we are able to focus on minimizing the equation shown in line 490, which is equivalent to g(f, A) stated in the main algorithm. However it is unclear how to efficiently solve g(f, A) for any arbitrary nonlinear function.
>
> In addition, our algorithm is based on previous passive learning results whose assumption on $\phi_W$ is linear.  Estimating nonlinear $\phi_W$ may introduce extra errors both from practical and theoretical perspectives. Therefore, we conservatively make this assumption.
>
> ### Q4: How do we decide the number of samples to select in each round of active sampling and what are some other key hyper-paramters?
>
> We agreed that, practically, some key parameters need to be adjusted either because of some idealistic assumptions or theory is pessimistically considering the worst case. Here are some main components:
>
> (1) **Number of samples $n_0$  in the warm-up stage**: One weakness is the warm-up length depends on $\bar{\kappa}$ which is the upper bound of the condition number of $B_W$. The exact value might be unknown to us in advance. Our current solution is to use some external knowledge to give an estimatation.
>
> (2) **Number of samples $n^j$ each round**:  In the theorem, $n^j$ depends on the model capacity. But in practice when using a neural network, it is well-known that its capacity is hard to estimate. Here, because we are using a shallow net, we simply replace $d_{\psi_W} d_{\psi_X}$ with the number of parameters in the net. How to choose with a more complicated model remains to be explored.
>
> (3) **Parameters in training oracle (learning rate schedule, optimization methods, epoch number)**:
> - Although we assume ERM during theoretical analysis, given the non-convexity of multi-task optimization, the training parameters of the optimizer need to be carefully selected . Here we use the initial warm-up stage result with a small fraction of held out validation data (½ of the total available few-shot target data)  to adjust the training parameters, Therefore, both passive and active algorithms use the exact same training oracles for fair comparison.
> - While here we do end-to-end training, some papers suggest more advanced training methods like alternatively minimizing the header and the representation layers may lead to better solutions.  Since the training method is not the main focus of this paper, we leave that for the future.
>
> Nevertheless, we want to emphasize the importance of theory that offers a qualitative tuning method that we can refer to. For example, (1) we should have three stages and the first stage shouldn’t depend on the target accuracy. (2) The second stage is only useful when $k << d_{\psi_W}$ (3) If using the second stage, the length should be $O( \text{lengthOfThirdStage}^{2/3})$, etc.
>
> ### Q5: The paper has dense notation dense but I appreciate the authors effort to unpack all the results ...
>
> A: Thanks for your suggestion, we will try to provide more explanation in the extra page.

---

> > ### Comment · Reviewer_7vYB · 2023-08-16
> > **Thank you for the detailed response!**
> >
> > I thank the authors for the detailed response to many of my questions and for incorporating suggestions. There are some hurdles to overcome to make the algorithm more practical, but the manuscript has novel ideas that warrant further exploration.

---

### Official Review · Reviewer_F2sU · 2023-07-07

**Soundness:** 3 good
**Presentation:** 3 good
**Contribution:** 3 good
**Rating:** 5
**Confidence:** 4

**Summary:**

This paper proposes a framework for active representation learning that allows learners to optimally choose which source tasks to sample from. The framework covers both task-aware and task-agnostic settings and is compatible with deep representation learning practices. The paper provides several instantiations of the framework, including a bilinear model and a neural network model. The authors demonstrate the effectiveness of their approach on a variety of simulated robotic tasks, showing that their method outperforms existing methods in terms of sample efficiency and generalization to new tasks. Overall, the paper's contributions include a novel framework for active representation learning, several instantiations of the framework, and empirical results demonstrating the effectiveness of the approach.

**Strengths:**

The paper proposes a novel framework for active representation learning that allows learners to optimally choose which source tasks to sample from. This framework covers both task-aware and task-agnostic settings and is compatible with deep representation learning practices. It considers a more general setting where tasks are parameterized in a vector space, allowing for more effective leveraging of similarities between tasks.

The paper provides a thorough theoretical analysis of the proposed framework, including sample complexity bounds and convergence guarantees. The paper is well-written and organized, with clear explanations of the proposed framework and its instantiations.

The proposed framework has broad applicability in robotics and other domains where there are multiple related tasks to be learned.

**Weaknesses:**

The manuscript would be considerably enhanced by the inclusion of more comprehensive empirical assessments, specifically pertaining to real-world robotic assignments, as opposed to merely simulated tasks. Such addition would serve to confirm the applicability and efficacy of the suggested method in a broader, more realistic context.

Further elucidation through an in-depth comparative study with established methodologies, notably those tailored towards active representation learning, would aid in comprehending the proposed method's advantages and shortcomings in relation to contemporary state-of-the-art techniques.

Moreover, a thorough discourse on the constraints inherent in the proposed method is warranted, especially considering the presumptions regarding the task space and task distribution. An exposition of this nature would lend itself to an improved understanding of the circumstances wherein the approach proves most efficient, as well as areas of potential inapplicability.

**Questions:**

Please see the Weakness.

**Limitations:**

Please see the Weakness.

---

> ### Author Rebuttal · Authors · 2023-08-09
>
> ### Q1: The manuscript would be considerably enhanced by the inclusion of more comprehensive empirical assessments, specifically pertaining to real-world robotic assignments, as opposed to merely simulated tasks. Such addition would serve to confirm the applicability and efficacy of the suggested method in a broader, more realistic context.
>
> Yes, a real-world robotics data collection driven by our algorithm and more complicated robotic tasks  is definitely the next step. But this paper mainly focuses on providing a general framework with theoretical guarantees, supported by some preliminary experiments. Notably, in order to narrow the gap between simulation and the real-word, here we provide the *real-world* drone flight dataset (line 294) in the discrete task setting. Note that such a dataset is pre-collected and contains a finite number of tasks. The experiment on a real-world + continuous environment would require an interactive data collection procedure in the real world, and is left as future research directions.
>
> ### Q2: Further elucidation through an in-depth comparative study with established methodologies, notably those tailored towards active representation learning, would aid in comprehending the proposed method's advantages and shortcomings in relation to contemporary state-of-the-art techniques.
>
> We apologize for this confusion.  As we discussed in "scope" and "literature review" section in our responses to all, the directly relevant previous work in this area is very limited. Before we submit this paper, only [1] considers this problem in a discrete setting. Later there is a follow-up paper [2] with improved results.
>
> Firstly, because those previous works are designed for discrete tasks, they are strictly and provably worse than our algorithm in most standard scenarios. We discuss this in Line 263-268. Due to the rigorous theoretical guarantees, we believe there is no need to repeat the same idea with experiments.
>
> Secondly, under the discrete space setting (where each $w$ is a one-hot vector) with a single target, previous works can be seen as a computationally efficient solution under our general framework. That is,  That is, the solution q from  [2]
>
> $v = argmin_v ||v||_1 \quad s.t. \hat{B}_W^\text{source} v =\hat{B}_W^\text{target} w_0$
>
> $q = v/||v||_1$
>
> can be seen as an efficient approximate solution of our proposed in line 12,
>
> $
> argmin_q  (\hat{B}_W^\text{target} w_0)^\top (\hat{B}_W^\text{source} Q (\hat{B}_W^\text{source})^\top)^{-1}  (\hat{B}_W^\text{target} w_0)
> $,
> where $Q = diag(q)$ and $q$ is the distribution over all source tasks.
>
> That is to say, we expect that, under this specific setting, our algorithm should be reduced to previous work. In fact, this is exactly what we do in our experiment: drone, where tasks are discrete, as stated line 757 -762.
>
>
> ### Q3: Moreover, a thorough discourse on the constraints inherent in the proposed method is warranted, especially considering the presumptions regarding the task space and task distribution. An exposition of this nature would lend itself to an improved understanding of the circumstances wherein the approach proves most efficient, as well as areas of potential inapplicability.
>
> A: Please refer to the “limitation” section in our response to all reviewers. Notably, this is the first result in the continuous space setting so the work we can compare with is relatively limited.

---

> > ### Comment · Area_Chair_Yvrj · 2023-08-18
> >
> > Dear Authors,
> >
> > The reviewer did not acknowledge your response, so I am joining the discussion. I have read the review and rebuttal and have no further questions (Q2 is shared with another review).
> >
> > Kind regards, Your AC

---

### Official Review · Reviewer_hbuX · 2023-07-10

**Soundness:** 2 fair
**Presentation:** 4 excellent
**Contribution:** 4 excellent
**Rating:** 6
**Confidence:** 1

**Summary:**

The paper studies active representation learning under the assumption that the source and target tasks are parameterized by a vector space (instead of a discrete set of tasks). The proposed algorithm works by first finding a subspace of the task space that spans the representation space and then iterates between refining the subspace and estimating for the target. The theoretical analysis shows that under certain assumptions, their active approach can achieve a better sample complexity than a passive one.

**Strengths:**

- The paper explores an understudied problem statement in active learning and theoretically demonstrates its importance in comparison to less expressive active learning problem statements, which could have a large impact on future work on the applicability of active learning.
- The paper is organized and the theory sections are well described.

**Weaknesses:**

- The problem statement and motivation were not described precisely or concisely. E.g., it's not clear to me why robotics is the only target for this work. The methods section was robotics-agnostic, but the introduction uses robotics as the motivating example and the paper is titled as a robotics paper.
- The experiment section lacks a comparison to baselines. E.g., a natural baseline would be to compare against an active learning approach that does assume a discrete source and task space. This would help show that the problem statement introduced is important empirically as well as theoretically. For this reason, I am recommending weak reject.

- Nit: The methods section also appeared to be accomplishing a lot of things at once---demonstrating the necessity of the continuous task space, showing the improvement in sample complexity over the passive approach, and minimizing expected excess risk---and it's not clear to the readers how these contributions relate or should be prioritized in our minds.
- Nit: the text on the LHS of figure 1 is cut off.

**Questions:**

Why do the Pendulum metrics only include test loss and control error? I would expect to see a measure of success or some proxy for it (e.g., average reward).

---

> ### Author Rebuttal · Authors · 2023-08-09
>
> ### Q1: The problem statement and motivation were not described precisely or concisely. E.g., it's not clear to me why robotics is the only target for this work. The methods section was robotics-agnostic, but the introduction uses robotics as the motivating example and the paper is titled as a robotics paper.
>
> Since one of our contributions is to provide a general framework, it is true that this problem may work for other tasks beyond robotics as long as the task satisfies our problem statement. But we want to emphasize that one of the key improvements is to show the effectiveness of our proposed algorithm in the continuous and infinite parameter space, which is not covered by previous work. Therefore, robotics (or cyber physical systems in general) is one of the most natural applications under such a setting.
>
> There are two specific reasons to choose robotics : 1) Robotics naturally need to deal with continuous physical environments like the wind speed, mass of pendulum, so we don’t need to “make up” those tasks. 2) Reducing the number of sources in robotics is a real demand, because switching between tasks/environments is much more expensive than collecting data in a fixed task/environment, and there are infinitely many tasks/environments to select from.
>
> We also notice that some other domains may satisfy these properties, such as some particular vision or graphics tasks. We are excited to implement our framework in other domains in the future.
>
> ### Q2: The experiment section lacks a comparison to baselines. E.g., a natural baseline would be to compare against an active learning approach that does assume a discrete source and task space. This would help show that the problem statement introduced is important empirically as well as theoretically. For this reason, I am recommending weak reject.
>
> We apologize for this confusion. As we discussed in the responses to all, the directly relevant previous work in this area is very limited. Before we submit this paper, only [1] considers this problem in a discrete setting. Later there is a follow-up paper [2] with improved results.
>
> Firstly, because those previous works are designed for discrete tasks, they are strictly worse than our algorithm in most standard scenarios. We show it with rigourious theoretical analysis and discuss this comparison in Line 263-268.
>
> Secondly, under the discrete space setting (where each $w$ is a one-hot vector) with a single target, previous works can be seen as a computationally efficient solution under our general framework. That is,  That is, the solution q from  [2]
>
> $v = argmin_v ||v||_1 \quad s.t. \hat{B}_W^\text{source} v =\hat{B}_W^\text{target} w_0$
>
> $q = v/||v||_1$
>
> can be seen as an efficient approximate solution of our proposed in line 12,
>
> $
> argmin_q  (\hat{B}_W^\text{target} w_0)^\top (\hat{B}_W^\text{source} Q (\hat{B}_W^\text{source})^\top)^{-1}  (\hat{B}_W^\text{target} w_0)
> $,
> where $Q = diag(q)$ and $q$ is the distribution over all source tasks.
>
> That is to say, we expect that, under this specific setting, our algorithm should be reduced to previous work. In fact, this is exactly what we do in our experiment: drone, where tasks are discrete, as stated line 757 -762.
>
>
>
> ### Q3: Nit: The methods section also appeared to be accomplishing a lot of things at once---demonstrating the necessity of the continuous task space, showing the improvement in sample complexity over the passive approach, and minimizing expected excess risk---and it's not clear to the readers how these contributions relate or should be prioritized in our minds.
>
> We are sorry for the confusion and will try to explain the logic here. The goal for our paper is to minimize the expected excess risk using as few source tasks and source samples as possible. To give a theoretical guarantee with explanations, Section 4 firstly provides an upper bound for the expected excess risk in terms of the sample complexity in Theorem 4.1. And then we explain this upper bound by showing that it is better to both the passive one (start from 243) and the previous active one (start from line 163). The comparison with the previous active one, consequently, implies the necessity of considering the continuous task – Because if we reduce the problem into discrete setting by random sample basis from the whole task space and apply previous AL approach, then we will get worse results. We will try to rewrite section 4 can give a brief summary on its structure. Please refer to the “scope” section in our responses to all reviewers for details and we will try to add those explanations in the extra page.
>
> ### Q4: Why do the Pendulum metrics only include test loss and control error? I would expect to see a measure of success or some proxy for it (e.g., average reward).
> The control error in this setting is the deviation from upright position, which is equivalent to negative reward https://www.gymlibrary.dev/environments/classic_control/pendulum/ [17].
>
> [17] Brockman, Greg, et al. "Openai gym." arXiv preprint arXiv:1606.01540 (2016).

---

> > ### Comment · Area_Chair_Yvrj · 2023-08-18
> >
> > Dear Authors,
> >
> > The reviewer did not acknowledge your response, so I am joining the discussion. I have read the review and rebuttal. I have a follow up question about Q2: I understand that [2] can be seen as an efficient approximate solution to your approach. Yet, would a comparison be possible? With the above, I would expect your method to be more accurate.
> >
> > Kind regards, Your AC

---

> > > ### Author Response · Authors · 2023-08-20
> > > **Relation between our work and [1][2] -- why use an efficient approximate solution**
> > >
> > > We thank AC for giving comments. We are sorry for the confusion on using the word “approximate”.
> > >
> > > Theoretically, we expect that [2] yields the same order result as our method, but we say it is an approximation because they are not rigorously equivalent. Motivated by this theoretical relationship, we expect that, empirically, a well-tuned version 2 should be very similar to our method in the discrete single target case. (But of course, we are focusing on more general case)
> > >
> > > Following is the detailed theoretical analysis on the relationship between our methods and [1][2].
> > >
> > > In this paper, we aim to minimize.
> > > $
> > > (\hat{B}_W^\text{target} w_0)^\top (\hat{B}_W^\text{source} Q (\hat{B}_W^\text{source})^\top)^{-1}  (\hat{B}_W^\text{target} w_0), \text{ where } \sum_i q_i = 1 \text{    [eq. 1]}
> > > $
> > >
> > > which can be equivalently written as
> > >
> > > $
> > > \text{trace} (V^\top Q V)^{-1} (V^\top uu^\top V) = \sum_i q_i^{-1} v_i^2
> > > $
> > >
> > > where $V$ comes from the eigendecomposition of $\hat{B}_W^\text{source}= UD V^\top$ and $u$ can be anything satisfying $ \hat{B}_W^\text{source} v =\hat{B}_W^\text{target} w_0$.
> > >
> > > Therefore, if we assume $q_i = |u_i|^\alpha / \sum_i |u_i|^\alpha$ for $\alpha > 0$, then [1] is equivalent to choosing $\alpha = 2$ and [2] is equivalent to  choosing $\alpha = 1$. And it is easy to see that $\alpha = 1$  is the optimal solution.
> > >
> > > Here we conservatively say it is an approximation because we didn’t prove that $q_i = |u_i|^\alpha / \sum_i |u_i|^\alpha$ is a right assumption to solve exact $q$, but we conjecture that the exact solution of eq.1 is very close to [2]. In addition, if we want to directly solve eq.1, we still need to use some approximation algorithms like Frankle-Wofle while implementing [2] can use existing optimized libraries with lasso modules. Due to these two reasons, we believe following a well-tuned methods in [2] should be a good choice in the discrete single target case
> > >
> > > **To summarize**, our general framework not only recovers the best result from the previous paper in this special case, but moreover gives strict improvements in the more general case (multi-targets, continuous space). Moreover, we want to emphasize that, although having similar results, [1][2] using quite different analysis techniques that are unclear how to extend the general setting. That is to say, as a side product, our work also provides novel and maybe more clear theoretical techniques to analyze such problems. For example, using our framework, it can be straight to discover that [2] is better than [1].
> > >
> > > We will add this detailed comparison in the modified version.

---

### Official Review · Reviewer_DHY3 · 2023-07-25

**Soundness:** 3 good
**Presentation:** 3 good
**Contribution:** 2 fair
**Rating:** 5
**Confidence:** 1

**Summary:**

The paper introduces and studies a theoretic and algorithmic framework for actively and optimally selecting the source tasks in multi-task learning, accommodating task-aware and task-agnostic settings.
The authors present cohesive theoretical guarantees, particularly demonstrated through the pure bilinear setting with the task or sample complexity being improved. They further empirically validate their meta-algorithm with task-parameter-wise non-linearity settings, particularly with emphasis on applications in robotics, significantly outperforming baselines.

**Strengths:**

The problem of active representation learning is interesting and important.
The proposed framework is novel and effective, demonstrated with both theoretical and empirical evidence.
The theoretical and experimental results are cohesive.

**Weaknesses:**

- While the method and algorithms seem sound, their explanation could be better with more interpretation and transitions.
- The experiment section (in the main manuscript) is not well-presented, e.g., lack of baselines/setting description as well as the interpretation and justifications of results.
- To my best assessment, the experiments are not systematically and thoroughly conducted.
**Minor:**
- The paper misses the conclusion, thorough literature review, and limitation parts

**Questions:**

See some questions in the *weakness* sections.

* What is the trade-off and limitations of the proposed algorithm compared to previous methods?
* When it comes to a single target task -- does the proposed method outperform the baseline?
* Given that the model applies to multiple target tasks, how does it compare to previous works when several target tasks are negatively correlated or have detrimental effects on each other?


**Limitations:**

The paper does not explicitly describe the limitations of the approach.
Please provide the limitations and the trade-off of the method.

---

> ### Author Rebuttal · Authors · 2023-08-09
>
> ### Q1: What is the trade-off and limitations of the proposed algorithm compared to previous methods?
>
> Please refer to our response to all reviewers. In particular, in the “limitation” section we discuss its limitations in both assumptions and the solutions.
>
> ### Q2: When it comes to a single target task -- does the proposed method outperform the baseline?
>
> In most continuous infinite parameter space: Our method is strictly better than reducing to discrete setting + previous methods [1][2], as discussed in line 163-171 and is also strictly better than the passive method (random sampling) as discussed in line 156-162. We shouw it by proving rigorous theoretical analysis.
>
> In the discrete parameter space, previous works can be seen as a computationally efficient solution under our general framework. That is, the solution q from  [2]
>
> $v = argmin_v ||v||_1 \quad s.t. \hat{B}_W^\text{source} v =\hat{B}_W^\text{target} w_0$
>
> $q = v/||v||_1$
>
> can be seen as an efficient approximate solution of our proposed in line 12,
>
> $
> argmin_q  (\hat{B}_W^\text{target} w_0)^\top (\hat{B}_W^\text{source} Q (\hat{B}_W^\text{source})^\top)^{-1}  (\hat{B}_W^\text{target} w_0)
> $,
> where $Q = diag(q)$ and $q$ is the distribution over all source tasks.
>
> That is to say, we expect that, under this specific setting, our algorithm should be reduced to previous work. In fact, this is exactly what we do in our experiment: drone, where tasks are discrete, as stated line 757 -762.
>
>
> ### Q3: Given that the model applies to multiple target tasks, how does it compare to previous works when several target tasks are negatively correlated or have detrimental effects on each other?
>
> This is beyond the scope of our paper. Our problem assumes that all the tasks (including target tasks) share the same representation as stated in eqn.(1), so they should *not* have detrimental effects on each other. It is a widely-used assumption in various kinds of pretraining/representation learning papers (see our “literature review ” section in the responses to all). In fact, we believe negatively learning correlated-targets simultaneously is an ill-posed problem, since in that case, collecting data and learning independently for each single target is intuitively the best solution.
>
> On the other hand, there are some existing works that consider negatively correlated *source* tasks as we mentioned in the literature review. In that case, they are mainly focused on rule-outing the harmful tasks instead of saving computational cost by selecting more useful tasks. Those works mostly have different assumptions and are beyond the scope of this paper.
>
> ### Weakness: While the method and algorithms seem sound, their explanation could be better with more interpretation and transitions. The experiment section (in the main manuscript) is not well-presented, e.g., lack of baselines/setting description as well as the interpretation and justifications of results. To my best assessment, the experiments are not systematically and thoroughly conducted. Minor: The paper misses the conclusion, thorough literature review, and limitation parts
>
> We are sorry for the confusion in presenting the method and algorithms and will try to add more explanation. For the experiment, due to the page limits, we selected the most important part in the main manuscript and left details and discussions in the appendix. We will leverage the extra page to add more intuitions and illustrations for the experimental results. For the limitation and literature part, please refer to our response to all.

---

> > ### Comment · Area_Chair_Yvrj · 2023-08-18
> >
> > Dear Authors,
> >
> > The reviewer did not acknowledge your response, so I am joining the discussion. I have read the review and rebuttal and have no further questions.
> >
> > Kind regards, Your AC

---

> > > ### Comment · Reviewer_DHY3 · 2023-08-18
> > > **Re: Rebuttal by Authors**
> > >
> > > Dear Authors,
> > >
> > > Thanks for your responses and clarification. I have read the other reviews and your answers in rebuttal. To my best assessment, I find the idea/methods sound and novel, but the manuscript still needs a proper revision of writing (and experiments) for a high-quality paper.

---

### Official Review · Reviewer_LcWv · 2023-07-26

**Soundness:** 2 fair
**Presentation:** 1 poor
**Contribution:** 2 fair
**Rating:** 1
**Confidence:** 1

**Summary:**

Which source task to sample from is an important issue in multi-task learning. This paper proposes a general and versatile algorithm and framework for active representation learning. The proposed algorithm can be adapted to arbitrary target and source task space, and can cover task-aware and task-agnostic settings. The experiments demonstrate the efficacy of the proposed algorithm on various datasets.

**Strengths:**

- Interesting and important topic.

**Weaknesses:**

- Incomplete contents.

- Poor writing and structure.

**Questions:**

None.

**Limitations:**

No limitations are addressed.

---

### Author Rebuttal · Authors · 2023-08-09

# Scope of this paper :

As pointed out by some reviewers, active learning, multi-task learning and representation learning is a broad area. However, our work is particularly focus on the following settings:

(1) We focus on assumptions that all the tasks share the same representation, which means learning from one task should not hurt another.

(2) We focus on data collection processing instead of the training process with a fixed candidate set of data.

(3) We focus on the task-wise active selection (i.e., actively choose the task $w$) instead of the sample-wise selection (i.e, actively choose the input $x$) as in many classical active learning works. This is particularly useful for robotics settings where switching between tasks/environments is much more expensive than collecting data in a fixed task/environment.

(4) While we propose a general framework, one of our major contributions is on the continuous infinite task space where many robotics applications lie, which is in contrast to many previous works that particularly focus on discrete CV/NLP tasks.

Notably, this is a relatively new task so the previous works are quite limited. In particular, there is no previous active representation learning work [1][2] under the *continuous, infinite parameter space*, so we are the first one who give the general formulation with *provable* algorithm and some preliminary  experiments. More specifically, [1] propose an algorithm under discrete single target setting and [2] further showing changing regularization function in [1] will lead to better result.

Since most of the paper focused on this novel problem formulation and theoretical proofs, we regard empirical as demonstrations to theoretical results when the idealist assumptions are violated. We left a more comprehensive empirical study and real-world experiment in the future.

[1]  "Active multi-task representation learning."

[2]  "Improved Active Multi-Task Representation Learning via Lasso."

# Literature review on related works

Even though the setting we consider is novel and there is limited previous work, we agree with some reviewers that it is useful to discuss broad multi-task learning and representation learning settings. Here we give a brief summary of other representation learning or multi-task papers that are related but different in some aspects we mentioned above. We will add those in the extra page in the revision.

**Multi-task with negative correlation** Some multi-task works [6,7,8,9] assume different tasks *don’t* share the same representation, so learning on one task may hurt another. They usually group similar tasks and assign an independent model to each group [6,7,8] or assign high weights on target-relevant sources [9]. The essential difference between those work and ours is that they assume a pass over the whole dataset is possible and aim to achieve the ultimate best performance, whereas we assume it is not (setting a large amount of experiment environment or maintaining a long time real data collection is costly). Consequently, they should not be considered as active.

**Passive Multi-task training/Meta learning**  While our paper focuses on data collection, some papers focus on the training process with some given dataset. For example, [9] mentioned above reweighting and joint-training all tasks. Another large topic in this scope is called “Meta-learning” [10,11,12], which usually focuses on more detailed updating methods. In conclusion, this line of works is parallel to our work, and all those methods can be regarded as a plug-in oracle in Line 5, 9, 12 in Algo 1 in our paper.

**Sample-wise data selection for representation learning** Classical pool-based active learning selects most informative data for a single task. Recently, some works [13,14,15] started to focus on selecting helpful data from a large corpus of web-scale for some known target task, where web-scale data could be seen as a mix of multi-task data without explicit “task” information.  Besides, those works usually focus on coarse labels and self-contrastive learning. Therefore, although they also aim to learn a presentation/pretrained model from non-target data, their detailed settings are quite different from ours.

**Discrete task space** Finally, most of the above papers consider discrete CV and nlp tasks while we consider robotic application that naturally has task parameters from the physical environment.

[3]  "Discovering structure in multiple learning tasks: The TC algorithm."

[4]  "Taskonomy: Disentangling task transfer learning."

[5]  "Task2vec: Task embedding for meta-learning."

[6]  "Which tasks should be learned together in multi-task learning?."

[7]  "Model Zoo: A Growing" Brain" That Learns Continually."

[8]  "Efficiently identifying task groupings for multi-task learning."

[9]  "Weighted training for cross-task learning."

[10] [11]  [12]  (skip here due to charater limits)

[13]  Data selection for language models via importance resampling.

[14]  DataComp: In search of the next generation of multimodal datasets.

[15] Cit: Curation in training for effective vision-language data. .

# Limitations of this setting

Some reviewers point out that the limitation of our works is not clear. Here we want to give a brief summary

**Limitations of our solutions** We have discussed the computational limitations with possible solutions in line 188 - 204, the suboptimality of generalization bound in benign case compared to the passive one in line 247-252, more theoretical limitations in Appendix C and more discussion on experiments in Appendix D

**Limitations from our setting** Comparing with related work, our solution

- is not robust to negatively correlated tasks.
- requires the existence of a good supervised (passive) multi-task representation learning oracle, which is a parallel to our study.
- only focuses on supervised-learning, instead of modern representation learning techniques (e.g. self-contrast) .

---

### Decision · Program_Chairs · 2023-09-21

**Decision:**

Accept (poster)

**Comment:**

The reviewers agreed to accept the paper on the grounds of introducing novel ideas for multi-task representation learning, sound theory with sample complexity bounds, and some limited experiments. The experimental side was the main concern, in particular the real world applicability. The discussion sufficiently addressed the weaknesses pointed out by the reviewers (especially comparisons with prior work), but the authors should include these updates to the camera ready.